

**ORCHIDEE-PEAT (revision 4596), a model for northern peatland**
**CO$_2$, water and energy fluxes on daily to annual scales**
Chunjing Qiu[1], Dan Zhu[1], Philippe Ciais[1], Bertrand Guenet[1], Gerhard Krinner[2], Shushi Peng[3],
Mika Aurela[4], Christian Bernhofer[5], Christian Brümmer[6], Syndonia Bret-Harte[7], Housen Chu[8],
Jiquan Chen[9], Ankur R Desai[10], Jiří Dušek[11], Eugénie S. Euskirchen[7], Krzysztof Fortuniak[12],
Lawrence B. Flanagan[13], Thomas Friborg[14], Mateusz Grygoruk[15], Sébastien Gogo[16,17,18], Thomas
Grünwald[5], Birger U. Hansen[14], David Holl[19], Elyn Humphreys[20], Miriam Hurkuck[6], Gerard
Kiely[21], Janina Klatt[22], Lars Kutzbach[19], Chloé Largeron[1,23], Fatima Laggoun-Défarge[16, 17, 18],
Magnus Lund[24], Peter M. Lafleur[25], Xuefei Li[26], Ivan Mammarella[26], Lutz Merbold[27], Mats B.
Nilsson[28], Janusz Olejnik[29,30], Mikaell Ottosson-Löfvenius[28], Walter Oechel[31], Frans-Jan W.
Parmentier[32,33], Matthias Peichl[28], Norbert Pirk[34], Olli Peltola[26], Włodzimierz Pawlak[12], Corinna
Rebmann[35], Daniel Rasse[36], Janne Rinne[34], Gaius Shaver[37], Hans Peter Schmid[22],
Matteo Sottocornola[38], Rainer Steinbrecher[22], Torsten Sachs[39], Marek Urbaniak[29], Donatella
Zona[30,40], Klaudia Ziemblinska[29]
1. Laboratoire des Sciences du Climat et de l'Environnement, UMR8212, CEA-CNRS-UVSQ
F-91191 Gif sur Yvette, France
2. CNRS, Université Grenoble Alpes, Institut de Géosciences de l'Environnement (IGE), F-38000
Grenoble, France
3. Department of Ecology, College of Urban and Environmental Sciences, Peking University,
100871 Beijing, China
4. Finnish Meteorological Institute, Climate Change Research, FI-00101 Helsinki, Finland
5. Technische Universität (TU) Dresden, Institute of Hydrology and Meteorology, Chair of
Meteorology, D-01062 Dresden, Germany
6. Thünen Institute of Climate-Smart Agriculture, Bundesallee 50, 38116 Braunschweig, Germany
7. Institute of Arctic Biology, University of Alaska Fairbanks, AK 99775 Fairbanks, USA
8. Department of Environmental Science, Policy, and Management, University of California,
Berkeley, 94720, CA, USA
9. Center for Global Change and Earth Observations, Michigan State University, East Lansing, MI
48823, USA
10. Department of Atmospheric and Oceanic Sciences, University of Wisconsin–Madison,
WI 53706 Madison, USA
11. Department of Matters and Energy Fluxes, Global Change Research Institute, Czech Academy
of Sciences, 603 00 Brno, Czech Republic
12. Department of Meteorology and Climatology, University of Łódź, Narutowicza 88, 90-139
Łódź, Poland
13. Department of Biological Sciences, University of Lethbridge, Lethbridge, T1K 3M4 Alberta,
Canada
14. Department of Geosciences and Natural Resource Management, University of Copenhagen,
Oester Voldgade 10, 1350 Copenhagen K, Denmark
15. Department of Hydraulic Engineering, Warsaw University of Life Sciences—SGGW,
Nowoursynowska 159, 02-776 Warszawa, Poland



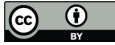

16. Université d'Orléans, ISTO, UMR 7327, 45071 Orléans, France
17. CNRS, ISTO, UMR 7327, 45071 Orléans, France
18. BRGM, ISTO, UMR 7327, BP 36009, 45060 Orléans, France
19. Institute of Soil Science, Center for Earth System Research and Sustainability (CEN),
47       Universität Hamburg, Germany

20. Department of Geography and Environmental Studies, Carleton University, K1S5B6 Ottawa,
49       Canada

21. Department of Civil and Environmental Engineering, University College Cork, Cork, Ireland
22. Karlsruhe Institute of Technology, Institute of Meteorology and Climate Research,
Atmospheric Environmental Research (IMK–IFU), 82467 Garmisch-Partenkirchen, Germany

23. CNRS and Univ. Grenoble Alpes, Institut de Géosciences de l'Environnement (IGE), 38000
Grenoble, France

24. Department of Bioscience, Arctic Research Centre, Aarhus University, 4000 Roskilde,
Denmark

25. School of the Environment - Geography, Trent University, Peterborough, Ontario, K9J 7B8,
Canada

26. Department of Physics, University of Helsinki, 00014 Helsinki, Finland
27. Mazingira Centre, International Livestock Research Institute (ILRI), 00100 Nairobi, Kenya
28. Department of Forest Ecology and Management, Swedish University of Agricultural Sciences,
S-90183 Umeå, Sweden

29. Department of Meteorology, Poznań University of Life Sciences, 60-649 Poznań, Poland
30. Department of Matter and Energy Fluxes, Global Change Research Center, AS CR, v.v.i.
Belidla 986/4a, 603 00 Brno, Czech Republic

31. Department of Biology, San Diego State University, CA 92182 San Diego,USA
32. The Arctic University of Norway, Institute for Arctic and Marine Biology, Postboks 6050
Langnes, 9037 Tromsø, Norway

33. Department of Geosciences, University of Oslo, Postboks 1022 Blindern, 0315, Oslo, Norway
34. Department of Physical Geography and Ecosystem Science, Lund University, 22362 Lund,
Sweden

35. UFZ-Helmholtz Centre for Environmental Research, 04318 Leipzig, Germany
36. Norwegian Institute of Bioeconomy Research, Oslo, Akershus, Norway
37. Marine Biological Laboratory, The Ecosystems Center, Woods Hole, 02543 Massachusetts,
USA

38. Department of Science, Waterford Institute of Technology, Waterford, Ireland
39. Helmholtz Centre Potsdam, GFZ German Research Centre for Geosciences, 14473 Potsdam,
Germany

40. Department of Animal and Plant Sciences, University of Sheffield, Western Bank, Sheffield
S10 2TN, UK


Correspondence to: Chunjing Qiu (chunjing.qiu@lsce.ipsl.fr)





## Abstract

Peatlands store substantial amount of carbon, are vulnerable to climate change. To predict the fate of carbon stored in peatlands, the complex interactions between water, peat and vegetations need more attention. This study describes a modified version of the ORCHIDEE land surface model for simulating the hydrology, surface energy and $CO_2$ fluxes of peatlands on daily to annual time scales. The model, referred to as ORCHIDEE-PEAT, includes a separate soil tile in each 0.5 ° grid-cell, defined from a global peatland map and identified with peat-specific soil hydraulic properties. Runoff from non-peat vegetation with a grid-cell containing a fraction of peat is routed to this peat soil tile, which maintains shallow water tables. The water table position separates oxic from anoxic decomposition. The model is evaluated against eddy-covariance (EC) observations from 30 northern peatland sites, with the maximum rate of carboxylation ($V_{cmax}$) being optimized at each site to match the peak of growing season gross primary productivity (GPP), derived from direct EC measurements. Regarding short-term variations from day to day, the model performance was good for the variations in GPP ($r^2 = 0.76$, Nash-Sutcliff modeling efficiency, MEF = 0.76), with lesser accuracy for latent heat fluxes (LE, $r^2 = 0.42$, MEF = 0.14) and Net ecosystem $CO_2$ exchange (NEE, $r^2 = 0.38$, MEF = 0.26). Seasonal variations in GPP, NEE and energy fluxes on monthly scales showed moderate to high $r^2$ values ranging from 0.57 to 0.86. For spatial across-sites gradients of annual mean GPP, NEE and LE, $r^2$ of 0.93, 0.27, and 0.71, respectively, were achieved. The water table variations are not well predicted ($r^2 < 0.1$), likely due to the uncertain water input to the peat from surrounding areas. However, when using the observed water table in the carbon module to define the fraction of oxic and anoxic decomposition instead of the modeled water table, ORCHIDEE-PEAT shows a small improvement in reproducing NEE. Moreover, we found a significant relationship between optimized $V_{cmax}$ and the latitude (temperature), which can better reflect the spatial gradients of annual NEE than using an average $V_{cmax}$ value. In a future version of ORCHIDEE-PEAT, the influences of water table on photosynthesis and depth-dependent influences of soil





temperature on respiration may be included.

## 1. Introduction

Peatlands cover only 3–5% of the Earth's land area, but store large amounts of soil
organic carbon. This carbon is primarily in the boreal and sub-arctic regions
(75–80%), while about 15% is in the tropical regions (Frolking et al., 2011; Page et al.,
2011). Current estimates of the northern peatland carbon stocks vary from 270 to 450
Pg C (Gorham, 1991; Turunen et al., 2002; Yu et al., 2010). Northern peat
accumulation occurred mainly during the Holocene, originating from plant litter
production exceeding decomposition in water-logged soil conditions, with low pH
and low temperatures (Parish et al., 2008). The future of the carbon stored in these
peatlands under the warmer environment and altered hydrological regimes is very
uncertain. Logically, higher $CO_2$ concentrations and elevated temperatures will
stimulate higher carbon uptake due to longer growing seasons and higher
photosynthetic rates (Aurela et al., 2004; Adkinson et al., 2011). However, the
accumulation is also coupled with a high evaporative demand that will lower the
ground water table, resulting in increased heterotrophic respiration rates (i.e., carbon
loss) (Mertens et al., 2001; Sulman et al., 2009; Adkinson et al., 2011). In addition to
the above potential climatic influences, other natural and anthropogenic disturbance
(permafrost melt, drainage, fires, etc.) can also play a role in determining the future
carbon balance of these vulnerable ecosystems (Turetsky et al., 2002; Parish et al.,
2008). Drainage and fires have particularly important impacts on the carbon balance
of the tropical peatlands (Page et al., 2002; Hooijer et al., 2010).
A number of peat carbon models have been reported in the literature. For example,
Frolking et al. (2010) created the Holocene Peat Model (HPM) which includes
feedbacks between plant communities, water table, peat properties, and peat
decomposition. This model was applied at Mer Bleue bog in southern Canada and
validated with data from peat-core observations. HPM is a long-term peat
accumulation model that works at an annual time step but cannot simulate seasonal





variations of key water processes in peatland. Wania et al. (2009a, 2009b) integrated
peatlands and permafrost into the Lund-Potsdam-Jena model (LPJ-WHy), in their
model, the upper 0.3 m of peatland soils (the acrotelm) experiences a fluctuating
water table and the underlying layer (the catotelm) is inundated permanently. A
constant soil moisture modifier (0.35) was used to reduce acrotelm decomposition.
Spahni et al. (2013) adopted and improved LPJ-Why to take into account the effects
of varying water table depth on acrotelm decomposition rates using a weighted
average of the aerobic and anaerobic respiration modifier, and implemented a
dynamic nitrogen cycle. In the dynamic global vegetation model (DGVM)
CLIMBER2-LPJ, Kleinen et al. (2012) determined the fraction of oxic decomposition
in the acrotelm by comparing the water table position and the acrotelm height.
Chaudhary et al. (2016, 2017) included a dynamic multi-layer peat accumulation
functionality in a customized Arctic version of the Lund-Potsdam-Jena General
Ecosystem Simulator (LPJ-GUESS). In their approach, new layers of litter were
added at the top of the soil every year, and the remaining litter mass after
decomposition was treated as a new individual peat layer from the first day of the
following year. The decomposition rate of peat, modulated by temperature and
moisture, declined over time. In these four peatland models, the water table depth is
calculated from a bucket model. In the context of Earth System Modeling, the land
surface is tend to be represented by several multi-layer schemes, such as multi-layer
plant canopy and root, multi-layer snow, multi-level soil carbon and energy budgets
(Best et al., 2011; Mcgrath et al., 2016; Zhu et al., 2016). To model peatlands
consistently in land surface models, a multi-layer soil hydrology scheme is needed.
Meanwhile, a more physics-based multi-layer scheme can provide more prognostic
power in predicting peatland water table dynamics.
In this study, we presented the results of the development of a peat hydrology and
carbon model in the ORCHIDEE land surface scheme, focusing on the water table
dynamics and its effects on the energy budgets, and on carbon decomposition
occurring within the oxic and the water-saturated part of the peat profile. $CH_4$ fluxes





and DOC loss through runoff are important components of the carbon balance of a
peatland (Chu et al., 2014; Olefeldt et al., 2012), but are not included in this study.
The originality of this new peat model is that it is incorporated consistently into the
land surface scheme in order to conserve water, carbon and energy at scales going
from local sites to grid-based large-scale applications in an Earth System Modeling
context. The model structure and equations are described in Sect. 2, and its evaluation
against water table depth, energy and $CO_2$ fluxes measured in 30 northern peat sites is
presented in Sect. 3.

**2. Model description**
**2.1 General structure of the model**
The ORCHIDEE land surface model simulates biophysical processes of rainfall
interception, soil water transport, latent (LE) and sensible (H) heat fluxes, heat
diffusion in the soil, and photosynthesis on a 30-min time step (Ducoudré et al., 1993).
Carbon cycle processes such as carbon allocation, respiration, mortality, litter and soil
carbon dynamics, are simulated on a daily time step (Krinner et al., 2005).
ORCHIDEE discretizes the vegetation into plant functional types (PFT): eight for
trees, two for natural C3 and C4 grasses, two for C3 and C4 crops, and one as
bare-soil type. Across the PFTs, plants are described with the same equations but
different parameter values, except for leaf onset and senescence that follow
PFT-specific equations (Botta et al., 2000). In grid-based simulations, PFTs are
grouped into three soil tiles: one with bare soil, one with all tree PFTs, and one with
all short vegetation. The water budget of each soil tile is calculated independently.
The version of ORCHIDEE implemented in this study uses the same (dominant) soil
texture for all the soil tiles of a grid cell to define the reference saturated hydraulic
conductivity ($K_{s\text{-}ref}$), and the saturated and residual volumetric water contents ($\theta_s$, $\theta_r$).
Dominant soil textural classes are taken from the Zobler's soil texture map (Zobler,
1986) at 1° resolution. The original five soil textures (fine, medium-fine, medium,
medium-coarse, coarse) in Zobler's map are reduced to three (fine, medium, coarse),



by grouping the medium-fine, medium, and medium-coarse into one class.
Hydrological parameters of the three dominant soil textures are taken from Carsel and
Parrish (1988) (Table 1).
Each soil tile in ORCHIDEE has eleven vertical layers (with a total depth of 2.0 m)
with exponentially coarser vertical resolution (Fig. 1). The Fokker-Planck equation is
used to describe the vertical diffusion of water in the soil. The Mualem (1976) - Van
Genuchten (1980) model (Eq. 1 and 2) is used to define the hydraulic conductivity (K,
m s$^{-1}$) and diffusivity (D, m$^2$ s$^{-1}$) as a function of volumetric water content ($\theta$, m$^3$m$^{-3}$):
$$K(\theta) = K_s \sqrt{\theta_f} (1 - (1 - \theta_f^{1/m})^m)^2 \quad , \tag{1}$$

$$D(\theta) = \frac{(1-m)K(\theta)}{\alpha m} \frac{1}{\theta - \theta_r} \theta_f^{-\frac{1}{m}} (\theta_f^{-\frac{1}{m}} - 1)^{-m} \quad , \tag{2}$$

where $\theta$ is the volumetric water content (m$^3$ m$^{-3}$), $\theta_s$ is the saturated water content (m$^3$
m$^{-3}$), $\theta_r$ is the residual water content (m$^3$ m$^{-3}$), $\theta_f$ is the relative water content and is
calculated as $\theta_f = \frac{\theta - \theta_r}{\theta_s - \theta_r}$ , $K_s$ is the saturated hydraulic conductivity (m s$^{-1}$), $\alpha$ is the
inverse of the air entry suction (m$^{-1}$), and $m$ is a dimensionless parameter.
Following Orgeval (2006) and Orgeval et al. (2008), $K_s$ exponentially decreases with
soil depth (z) below $z_{lim} = 30$ cm ($F_d(z)$), while a root-fracturing factor increases $K_s$
where roots are denser ($F_{root}(z)$):
$$K_s(z) = K_{s-ref} \times F_d(z) \times F_{root}(z) \quad , \tag{3}$$

with $F_d(z) = \min(\max(\exp(-f(z - z_{lim})), 0.1), 1)$, $F_{root}(z) = \prod_{j \epsilon c} \max(1, (\frac{K_s^{max}}{K_{s-ref}})^{\frac{1 - \alpha_j z}{2}})^{f_j}$ ,
where $K_{s-ref}$ is the reference top-soil saturated hydraulic conductivity determined by
soil texture (m s$^{-1}$), $K_s^{max}$ is the value of the coarser (sandy) texture and equals 8.25
$\times$ 10$^{-5}$ m s$^{-1}$, $\alpha_j$ is a root profile decay factor for PFT $j$ with a coverage fraction f$_j$, and
$c$ is the soil tile to which PFT $j$ was assigned to.

**2.2 Modifications in ORCHIDEE-PEAT**
To simulate peat, we: 1) modified the parameters of plants growing on peat, 2) added
a new peat soil tile with specific peat soil hydraulic properties, and 3) changed the





decomposition of peat carbon as being controlled by saturated conditions, through the
modeled water table (WT).
**Modified peat plant parameters.** As a response to the unique stress conditions in
peatlands (i.e., oxygen deficit, nutrient limitation), peatland vegetation has shallow
and extensive root systems (Boutin and Keddy, 1993; Iversen et al., 2015). In this
study, a C3 grass peatland PFT with a rooting depth of 30 cm implemented by
Largeron et al. (2017) was used. The maximum rate of carboxylation ($V_{cmax}$) typically
varies across peat sites (Rennermalm et al., 2005; Bubier et al., 2011), and further
varies with leaf nitrogen, phosphorus content, and specific leaf area (Wright et al.,
2004; Walker et al., 2014). For instance, $V_{cmax}$ value for *Sphagnum* at the Old Black
Spruce site (53.985 °N, 105.12 °W) in Canada were 5, 14 and 6 $\mu mol\ m^{-2}\ s^{-1}$ during
spring, summer and autumn respectively, while that for *Pleurozium* were 7, 5, and 7
$\mu mol\ m^{-2}\ s^{-1}$ (Williams and Flanagan, 1998). Bui (2013) conducted a fertilization
experiment at the Mer Bleue Bog (Canada, 45.41 °N, 75.52 °W) on the dominant
ericaceous shrub and reported that $V_{cmax}$ values ranged between 6 and 179 $\mu mol\ m^{-2}$
$s^{-1}$, with significantly higher $V_{cmax}$ values after addition of nitrogen (6.4 g N $m^{-2}\ year^{-1}$)
at 20 times the growing season ambient wet N deposition rate with or without
phosphorus (P) and potassium (K). In this study, we used a default $V_{cmax}$ value of 16
$\mu mol\ m^{-2}\ s^{-1}$ for peat PFT, following a literature survey by Largeron et al. (2017).
Later (Sect. 4.1), we calibrated $V_{cmax}$ at each site so that modeled peak gross primary
production (GPP) matched peak values derived from direct EC measurements, and
then regressed this adjusted $V_{cmax}$ value with environmental and climate variables. We
note that this adjustment of $V_{cmax}$ may over- or under-compensate for biases in other
model parameters that impact maximum GPP, such as leaf area index (LAI), specific
leaf area (SLA), canopy light absorption parameters, water and temperature stresses.
**Peat-specific soils hydraulics.** Peatlands generally occur in flat areas that are poorly
drained and/or receive runoff and sub-surface water from the surrounding landscape
(Graniero and Price, 1999). The low permeability catotelm peat layer is permanently
saturated. In ORCHIDEE-PEAT, the new soil tile added in a grid cell to represent

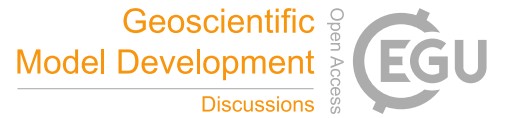



peatland as a landscape element was assumed to receive surface runoff from the other
three soil tiles (bare soil, trees, grasses) and has a drainage flux reduced to zero
(Largeron et al., 2017). Further, considering that the water table of a peatland can rise
above the ground surface, an above surface water reservoir with a maximum height of
10 cm was added (Fig. 1b). This reservoir loses water to rivers when filled, and
re-infiltrates into the peat soil on the next time step (Largeron et al., 2017). We
verified that the measured standing water remained below 10 cm above the soil
surface at 16 out of 20 northern peat sites where water table depth was recorded in
this study (Table S1). The four exceptions were Winous Point North Marsh
(US-WPT), Himmelmoor (DE-Hmm), an Alaska fen (US-Fen) and an Alaska bog
(US-Bog), where observed water tables reached up to 77 cm, 39 cm, 46 cm and 34 cm
above the soil surface, respectively.
Peat soils cannot be described with any of the mineral soil textures used for other
tiles (Table 1) because the low bulk density and high porosity increase the downward
water percolation (Rezanezhad et al., 2016). There is a large variability of observed
peat saturated hydraulic conductivity (K) and diffusivity (D) in space, depth and time.
This is partly related to the degree of decomposition and compression of organic
matter (Gnatowski et al.,2010). Morris et al. (2015) reported near-surface saturated
hydraulic conductivities (K) of $2.69 \times 10^{-2}$ m s$^{-1}$ to $7.16 \times 10^{-6}$ m s$^{-1}$ in bogs.
Gnatowski et al. (2010) measured values of $5 \times 10^{-6}$ m s$^{-1}$ in a moss-covered peat,
which is two orders of magnitude larger than in a woody peat ($5.56 \times 10^{-8}$ m s$^{-1}$).
Peat hydraulic parameters values used in this study were applied after (Largeron et al.,
2017), based on Letts et al. (2000) and Dawson (2006) (Table 1). The peat saturated
hydraulic conductivity value of $2.45 \times 10^{-5}$ m s$^{-1}$ is comparable to the harmonic
mean value ($6 \times 10^{-5}$ m s$^{-1}$) of Morris et al. (2015). The values of the other Van
Genuchten parameters for peat (Table 1) are similar to those employed in other
peatland models (Wania et al., 2009a; Wu et al., 2016).



The peatland water table depth (WT, cm) is diagnosed by summing water heights in
the eleven soil layers, calculated from the relative water content (Largeron et al.,

2017):

$$WT = H_{tot} - \sum_{i=1}^{11}(\theta_{fi} * dz_i) - H_{ab}; \text{ with } \theta_{fi} = \frac{\theta_i - \theta_r}{\theta_s - \theta_r},$$    (4)
where $\theta_{fi}$ is the relative volumetric water content of the $i^{th}$ soil layer, $\theta_s$ is the saturated
water content ($m^3$ $m^{-3}$), $\theta_r$ is the residual water content ($m^3$ $m^{-3}$), $dz_i$ is the distance
between node $i$-$1$ and node $i$ (Fig. 1, m), $H_{tot}$ is the total soil column height being
fixed to 2.0 m, and $H_{ab}$ is the height of the water reservoir above soil surface (m).
Thus, when the water table is above the surface, the modeled WT takes negative
values.
**Decomposition of peat carbon controlled by water saturation.** In the standard
version of ORCHIDEE, plant litter carbon is added to two litter pools: the metabolic
and the structural pool. Decomposed litter carbon from these two pools is then
distributed into three soil carbon pools: the active, slow and passive pool (Fig. S1),
similar to the CENTURY model (Parton et al., 1988). Both temperature and moisture
functions are used to control soil carbon decomposition rates (Text S1).
The original decomposition equations are combined with a new module to account
for peat decomposition being controlled by water saturation, after Kleinen et al. (2012)
(Fig. S1). Specifically, carbon from decomposed litter pools is added to the acrotelm
carbon pool where it is then decomposed aerobically above the simulated water table,
and anaerobically below it. The permanently saturated deep catotelm carbon pool
receives a prescribed fraction (1.91% per year, Kleinen et al., 2012) of the acrotelm
carbon, and is decomposed anaerobically at a very slow rate ($3.35 \times 10^{-5}$ $yr^{-1}$,
Kleinen et al., 2012). Whereas acrotelm depth was fixed to 30 cm in some two-layer
peat decomposition models (Yurova et al., 2007; Wania et al., 2009a; Spahni et al.,
2013), we used the average of minimum summer water table position ($WT_{min}$) over
the observational period to demarcate the boundary between the acrotelm and the
catotelm at each site to take into account local site conditions. $WT_{min}$ values were





estimated based on current climate due to the lack of knowledge of initiation histories
of these sites. For the long-term carbon accumulation estimations, the Holocene
climate may be a better proxy since northern peatlands show peak initiation in the
early Holocene (Yu et al., 2010). By comparing the height of the acrotelm (Fig. S1)
with the WT depth, we derived the fraction of the acrotelm where carbon decomposes
under oxic ($\beta$) vs. anoxic conditions (1-$\beta$). Acrotelm height was calculated from
acrotelm carbon stock ($C_A$ in Eq. 5-7), acrotelm carbon fraction ($C_{f,A}$) and acrotelm
bulk density ($\rho_A$). Decomposition of peat carbon is then controlled by temperature and
parameterized as an exponential function $Q_{10}\exp((T-T_{ref})/10\,°C)$ with $Q_{10} = 2.0$ and $T_{ref}$
$= 30\,°C$ (Text S1). Soil carbon fluxes are given by:
$$F_{AC} = k_p C_A \quad , \tag{5}$$

$$R_{A,o} = \beta k_A C_A \tag{6}$$

$$R_{A,a} = (1 - \beta)\mu k_A C_A \quad , \tag{7}$$

$$R_C = k_C C_C \quad , \tag{8}$$

where $F_{AC}$ is the carbon flux from acrotelm to catotelm; $R_{A,o}$ is aerobically
decomposed acrotelm carbon; $R_{A,a}$ is anaerobically decomposed acrotelm carbon; $R_C$
is decomposed carbon in catotelm; $C_A$ is carbon stored in the acrotelm; $C_C$ is carbon
stored in the catotelm; and $\beta$ is the fraction of acrotelm under oxic conditions. A
10,100 years' spin-up was conducted to initialize peat depth at each site (Sect. 3.3).
Following the study of Kleinen et al. (2012), the catotelm formation rate $k_p = 1.91\times$
$10^{-2}\,yr^{-1}$, the acrotelm decomposition rate $k_A = 0.067\,yr^{-1}$, the catotelm decomposition
rate $k_C = 3.35 \times 10^{-5}\,yr^{-1}$, the ratio of anaerobic to aerobic $CO_2$ production $\mu = 0.35$,
carbon fraction in the acrotelm peat $C_{f,A} = 0.50$, the acrotelm density $\rho_A = 3.5 \times 10^4\,g$
$m^{-3}$, carbon fraction in the catotelm peat $C_{f,C} = 0.52$, the catotelm density $\rho_C = 9.1 \times$
$10^4\,g\,m^{-3}$.
In the following analysis, carbon fluxes are defined positive if upwards. Thus,





ecosystem respiration is positive, GPP is negative, and a negative NEE signifies the
uptake of $CO_2$ by the ecosystem.

**3.  Validation of ORCHIDEE-PEAT at northern hemisphere peatland**

**eddy-covariance sites**

**3.1 Sites description**
To evaluate the performance of ORCHIDEE-PEAT in simulating $CO_2$, water and
energy fluxes on daily to annual time scales among the peatlands, we compiled data
from 30 northern peatland sites where eddy-covariance data and physical variables
(water table, snow depth, soil temperature) were collected (Fig. 2, Table 2). These
sites are spread between the temperate to the arctic climate zones, and include nine
bogs and 18 fens. A marsh and two wet tundra sites (these two wet tundra sites are
neither a fen nor a bog, hereafter they are referred to as 'tundra') with a ~30–50 cm
thick organic layer are also included in this study. Among them, six sites are underlain
by permafrost and one site is in a thermokarst area. The peatland fractional cover in
the 0.5 ° grid cell containing each site is from the Yu et al. (2010) map (Fig. 2, Table 2).
A short description of all sites can be found in Supplementary Materials.

**3.2 Meteorological forcing data**
We ran the model for 29 different 0.5 ° grid cells corresponding to each peatland site
(US-Fen and US-Bog are in the same grid cell, but their local meteorological data was
different). Peatland fraction in each grid cell was prescribed from Yu et al. (2010),
adapted by Largeron et al. (2017) to be matched with a high-resolution land cover
map. For cells (15 out of 29) without peatland (Fig. 2, Table 2) in the large-scale map
from Yu et al. (2010), a mean peatland fraction of 22% was assigned.

Time series of half-hourly air temperature, wind speed, wind direction, long-wave

incoming radiation, short-wave incoming radiation, specific humidity, atmospheric
pressure, and precipitation were used to drive ORCHIDEE-PEAT. All mentioned
variables were from measurements made at each flux tower where $CO_2$ and energy





(latent heat (LE) and sensible heat (H)) fluxes, water table position, soil temperature,
and snow depth were recorded on a half-hourly time step. The linearly interpolated
6-hourly CRU-NCEP 0.5 ° global climate forcing dataset was used to fill the gaps. A
linear correction was applied to meteorological forcing variables (except precipitation)
in the CRU-NCEP dataset to match observations before gap-filling. For precipitation,
no correction was applied. At CA-Wp2 and CA-Wp3, meteorological forcing data
were measured only during the growing season, so CRU-NCEP data were linearly
corrected using relationships derived from the available data. For some sites, several
meteorological data were not measured, such as long-wave incoming radiation at
NO-And, atmospheric pressure, short-wave incoming radiation, and long-wave
incoming radiation at CZ-Wet. In these cases, CRU-NCEP data were used to fill the
gaps without correction.

**3.3 Model setup**
ORCHIDEE-PEAT was first spun-up for 10,100 years, forced by the preindustrial
atmospheric $CO_2$ concentration of 285 ppm, with repeated site-specific observational
meteorological fields, and present-day vegetation fractions for each site. In reality, the
climate changed through the Holocene, but since the initiation and climate history of
each site are unknown, we assumed a constant present-day climate condition and
peatland area. Thus this model is only suitable for simulating water, energy and $CO_2$
fluxes from peat on time scales ranging from days to decades. To accelerate the
spin-up, ORCHIDEE-PEAT was first run for 100 years to reach the equilibrium for
hydrology and soil thermal conditions, fast carbon pools and soil carbon input from
dead plants. Then, a sub-model simulating only soil carbon dynamics (with fixed
daily litter input from the previous simulation) was run for 10,000 years to accumulate
soil carbon. Peatlands can reach equilibrium only when the addition of carbon equals
carbon lost, which is attained on time scales of $10^4$ years (Clymo, 1984; Wania et al.,
2009b). The catotelm carbon pool in this study was still not fully equilibrated even
after 10,100 years due to the low carbon decomposition rate in this reservoir (3.35 ×





$10^{-5}$ $yr^{-1}$, Kleinen et al., 2012). The modeled peat carbon pool thus depends on the
time length of spin-up, which was fixed at 10,100 years. While in the real world, peat
age at some sites can be younger. For example, the sample from the second last 10 cm
peat segment at CA-Wp1 has an un-calibrated radiocarbon date of ~2200 years
(Flanagan and Syed, 2011). Since we focus on carbon and water fluxes on daily to
annual scales in this study, rather than on the simulation of peat carbon stocks, we
conducted a sensitivity analysis of modeled heterotrophic respiration to the length of
the spin-up, which shows only a slight increase of catotelm respiration with increasing
simulation time (Fig. S2). After the spin-up, transient simulations were conducted for
each site, forced by repeated site-specific climates and rising atmospheric $CO_2$
concentration during the period 1901-2015. Finally, the model outputs corresponding
to the respective measurement periods (all during 1999-2015) were compared to
observed time series for each site.

Two sets of simulations were conducted. In the first one (S1), soil water content

and water table position (WT) were modeled by ORCHIDEE-PEAT, and the WT was
used in the carbon module to define the fraction of oxic and anoxic decomposition in
the acrotelm. S1 was performed for all the 30 sites. In the second set (S2) of
simulations, we prescribed water table in the model to equal to observed values
($WT_{obs}$). That is, soil moisture at layers below the measured water table was
prescribed as saturated ( $\theta(z > WT_{obs}) = \theta_s$ ), while soil moisture above $WT_{obs}$ was
simulated. $WT_{obs}$ was further used in the carbon module in S2. S2 was performed only
for a subset of eight sites where at least two years of water table measurements were
available and where there were sufficient observations to gap-fill the $WT_{obs}$ time
series (Table 2). For these sites, the gaps of $WT_{obs}$ were filled with the mean value of
the same period from other years of measurement (Table S2). The simulation S2 was
designed to check if the model performance will improve (or deteriorate) when
prescribing WT exactly to its observed value, since WT is known to be a critical
variable impacting peat water and $CO_2$ exchange, and $CH_4$ emissions (Dušek et al.,
2009; Parmentier et al., 2011; Strack et al., 2006). Fixing the simulated water table to

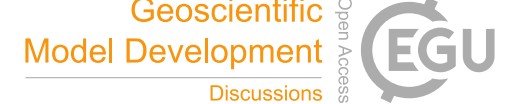



WT$_{obs}$ in S2 violated the water mass conservation of the model but allowed us to
evaluate the carbon module independently from the hydrological module biases.

**3.4 Measures for evaluating model performance**
Following Jung et al. (2011) and Tramontana et al. (2016), we used site-specific daily
means, annual means, seasonal variations and daily anomalies to evaluate the model
performance. For each site, seasonal variations are calculated by removing the annual
mean value from the mean seasonal cycle (averaged value for each month across all
available years), anomalies are calculated as the deviation of a daily flux value from
the corresponding mean seasonal cycle.

A series of measures were used to assess the model performance (Kobayashi and

Salam, 2000; Jung et al., 2011; Tramontana et al., 2016).

The root mean square deviation (RMSD) reports the model accuracy by measuring

the differences between simulation and observation.
$$\text{RMSD} = \sqrt{\frac{1}{n}\sum_{i=1}^{n}(x_i - y_i)^2} \quad , \qquad\qquad (9)$$
where $x_i$ is simulated variable, $y_i$ is measured variable, and $n$ is the number of
observations.
Two signals (SDSD and LCS) are discriminated from the mean squared deviation
(Kobayashi and Salam, 2000). The squared difference (SDSD) between the standard
deviation of the simulation (SD$_s$) and the measurement (SD$_m$) shows if the model can
reproduce the magnitude of fluctuation among the $n$ measurements.
$$\text{SDSD} = (\text{SD}_s - \text{SD}_m)^2; \text{ with } \text{SD}_s = \sqrt{\frac{1}{n}\sum_{i=1}^{n}(x_i - \bar{x})^2} \ , \text{SD}_m = \sqrt{\frac{1}{n}\sum_{i=1}^{n}(y_i - \bar{y})^2} \quad , \qquad (10)$$
where $\bar{x}$ is simulated mean value, $\bar{y}$ is measured mean value.
The lack of correlation weighted by the standard deviations (LCS) is a measure to
examine if the model reproduces the observed phase of variability.
$$\text{LCS} = 2\text{SD}_s\text{SD}_m(1 - r); \text{ with } r = \left[\frac{1}{n}\sum_{i=1}^{n}(x_i - \bar{x})(y_i - \bar{y})\right]/(\text{SD}_s\text{SD}_m) \quad ,$$

(11)

where $r$ is the Pearson's correlation coefficient.





The Nash-Sutcliff modeling efficiency (MEF) is used to indicate the predictive
accuracy of the model. MEF varies between negative infinity (–inf) and 1, an
efficiency of 1 indicates a perfect fit between simulations and observations; an
efficiency of 0 indicates the simulations are as accurate as the mean value of
observations; a negative MEF indicates that mean value of observations has greater
predictive power than the model.
$$\text{MEF} = 1 - \frac{\sum_{i=1}^{n}(x_i - y_i)^2}{\sum_{i=1}^{n}(y_i - \bar{y})^2},$$
(12)

**4. Results**
**4.1 Site-specific $V_{cmax}$ reduces errors in carbon simulations**
Out of the 30 sites, 22 sites provided observed daily GPP (separated from measured
NEE). The values of optimized $V_{cmax}$ at each site were listed in Table 3. The
optimized $V_{cmax}$ varied from 19 to 89 μmol m$^{-2}$ s$^{-1}$ (Table 3), with a mean value of 40
μmol m$^{-2}$ s$^{-1}$, which was higher than the default value (16 μmol m$^{-2}$ s$^{-1}$) fixed by
(Largeron et al., 2017).
Taylor diagrams were used to evaluate model results at these 22 sites (Fig. 3). The
model had the best performance for GPP, with the correlation coefficient between
simulated and observed GPP varied between 0.66 and 0.93 and all data points fell
within the 0.9 root mean square difference circle. Simulated water table depth had a
larger spread in correlation (0.16–0.82) and root mean square difference (0.4–4.0). We
found no significant patterns of model-data misfits among different peatland types
(fen, bog, others) or climate zones (temperate, boreal and arctic), as shown by
different shapes or colors of markers in Fig. 3.
For the 22 sites where GPP observations were available, the modeled GPP errors
were significantly reduced by optimizing $V_{cmax}$ at each site (Table 4). When a fixed
$V_{cmax}$ value (16 μmol m$^{-2}$ s$^{-1}$) was used, GPP was generally underestimated and
across-sites differences were not reproduced (Fig. S3, Table 4). Unsurprisingly,
neither the spatial nor the temporal variations of NEE were captured by the model
when using the fixed $V_{cmax}$ value (Fig. S3, Table 4). With site-specific $V_{cmax}$ values



(Site-by-site model performances are shown in Fig. S6 to S10 in Supplementary
Materials), the overall (all the daily data from all the 22 sites) performance of the
model was improved for GPP ($r^2$ = 0.76, MEF = 0.76), LE ($r^2$ = 0.42, MEF = 0.14),
NEE ($r^2$ = 0.38, MEF = 0.26) and sensible heat ($r^2$ = 0.24, MEF = -0.50) (Fig. 4, Table
4). Seasonal variations in carbon and energy fluxes were generally well captured by
the model ($r^2$ = 0.57 to 0.86). The spatial across-sites gradients of annual mean GPP,
NEE and LE were generally good, with $r^2$ of 0.93, 0.27, and 0.71 and RMSDs of 0.41
g C m$^{-2}$ day$^{-1}$, 0.60 g C m$^{-2}$ day$^{-1}$ and 9.85 W m$^{-2}$, respectively. The model
performance was poor for predicting daily anomalies of all fluxes, with $r^2 < 0.20$. For
both temporal and spatial variation, the MEF of the WT were negative, and $r^2$ smaller
than 0.10, indicating that the model had the lowest predictive capability for the WT.
Possible reasons for this could be: 1) peat management was not parameterized; i.e.,
the removal of beaver dams resulted in a decline of water level at US-Los; water level
at US-WPT, CZ-Wet and RU-Che were manipulated; 2) the model diagnosed all
peatland sites as fens by routing runoff from non-peatland areas into the peatland soil
tile, whereas in the real world, bogs are only fed by precipitation. In other words, we
included an extra water source for bogs than only rainfall; and 3) WT depends on
water input from surrounding non-peatland areas, and the peatland area fraction
derived from the map of Yu et al. (2010) could not represent the local area providing
water for fens; 4) for global applications, the effects of micro-relief cannot be
represented in the model, which has been proven to be an important regulator of the
local hydrology cycle (Gong et al., 2012; Shi et al., 2015).

To better understand the influence of the water table dynamics on NEE in the

model, we compared the second set of simulations (S2, the observed water table was
used in the carbon module to define the fraction of oxic and anoxic decomposition in
the acrotelm) with the first set (S1, the water table dynamics was calculated by the
model). ORCHIDEE-PEAT showed only a small improvement in reproducing NEE
when WT$_{obs}$ was used (Table 5). To illustrate this effect, we took the Lompolojänkkä
(FI-Lom) fen site as an example, in which WT was seriously underestimated (Fig. S8).



While modeled WT varied between 5 and 54 cm below the surface, $WT_{obs}$ was always
above the soil surface. Fig. 5a showed that in comparison to S1, there was no aerobic
respiration and larger anaerobic respiration in the acrotelm in S2. Due to the smaller
acrotelm respiration (aerobic + anaerobic) in S2, there was larger carbon input from
acrotelm to catotelm and consequently, there was more carbon accumulated in the
catotelm in S2. Thus, the catotelm respiration in S2 was greater than in S1 (Fig. 5c),
even though the catotelm respiration was very small. Because the growth of the
peatland vegetation was not constrained by water in the model, the simulated GPP
values were similar between S1 and S2 (Fig. 5a). With similar GPP but smaller soil
respiration (the acrotelm + the catotelm respiration), S2 simulations thus resulted in
more negative NEE values than S1 (higher net $CO_2$ uptake). Simulated leaf onset
occurred earlier than observed at Lompolojänkkä site, causing the ecosystem switched
from a carbon source to a carbon sink in May, while in observations the start of the
carbon uptake was later (Fig. 5b). Although the model reproduced a similar amplitude
of the observed NEE, the day-to-day variations of this flux were not captured (Fig. 6),
causing an overestimation (more negative values) of NEE in the warm period
(May-September).


**4.2 Relationship between optimized $V_{cmax}$ and meteorological variables**
Several uni-variate ANOVA models were used to explain the spatial gradient of
optimized $V_{cmax}$, explanatory variables including air temperature (T), precipitation (P),
net radiation (NET_RAD), water use efficiency (WUE), water balance (WB) and the
latitude (LAT). All explanatory variables were calculated as daily mean values during
the growing season. Water use efficiency (g C $m^{-2}$ $mm^{-1}$ $H_2O$) was calculated as the
ratio of GPP and evapotranspiration. Water balance (mm/day) was calculated as the
difference between precipitation and evapotranspiration.
There was no significant difference between optimized $V_{cmax}$ among peatland types
(fen vs bog, p = 0.16), climate zones (temperate vs boreal vs arctic, p = 0.17), or
dominant vegetation types (grasses and/or mosses dominated vs shrubs and/or trees





dominated, p = 0.67) (Fig. S4). However, we found a significant positive relationship
between $V_{cmax}$ and the growing season mean air temperature (Fig. S5, Table 6, $r^2$ =
0.19, p < 0.05) and a significant negative relationship between $V_{cmax}$ and the latitude
of the sites location (Fig. S5, Table 6, $r^2$ = 0.23, p < 0.05).
To verify the applicability of the empirical relationship found across sites between
optimized $V_{cmax}$ and the latitude (Fig. S5, slope = -0.92, intercept = 93.56, $r^2$ = 0.23, p
< 0.05), we used the seven sites where there were no GPP observations available
(US-Bes, DE-Hmm, US-Ics, PL-wet, SE-Sto, CA-Wp2 and CA-Wp3) as
cross-validated sites. We compared model performance in simulating NEE with $V_{cmax}$
being calculated according to the empirical relationship, and with $V_{cmax}$ being fixed to
its mean value of all 22 sites from Table 3 (40 μmol m$^{-2}$ s$^{-1}$). The model performance
in reproducing spatial gradients of NEE was improved when the $V_{cmax}$ values derived
from the empirical relationship were used (Fig. S11b, with RMSD reduced by 11%, $r^2$
increased from 0.20 to 0.38, and MEF increased from -0.04 to 0.17). This implies that
compared to a fixed $V_{cmax}$, the usage of $V_{cmax}$ value from the empirical relationship
can better capture spatial gradients of NEE. It is worth mentioning that the empirical
relationship was built on climate conditions from the last two decades (1999-2015),
and thus may change in the future when the climate changes.

**4.3 Soil temperature and a snow depth underestimation in the model**
For most of the sites, soil temperature was underestimated in winter and
overestimated in summer by our model (Fig. 7 and 8, results from sites DK-Nuf and
CA-Wp1). One possible reason for the underestimation of soil temperature in winter
is the underestimation of a snow depth (Fig. 9), since snow insulates the soil changing
thermal conditions in comparison to a snow-free surface. The underestimation of the
snow depth can be caused by the bias in snow processes of the model, such as
overestimation of a snow density and subsequently overestimation of snow
compaction, and/or overestimation of its sublimation. The insulation effects of the
moss layer and the top organic layer are not included in this study, which may explain





why soil temperature was overestimated in summer but underestimated in winter.

ORCHIDEE-PEAT calculates one energy budget for the vegetation and soil columns

in one grid cell. Key parameters used for solving the heat diffusion equations in the

soil, such as soil heat capacity and thermal conductivity, were prescribed by the

dominant soil texture in the grid cell (Gouttevin et al., 2012). Nevertheless, similar to

the case of hydrology module, the three default (coarse, medium, fine) soil textures

cannot represent thermal properties of a peat soil (Paavilainen and Päivänen, 1995;

Abu-Hamdeh and Reeder, 2000).

**5. Discussion**

ORCHIDEE-PEAT grouped various peatland vegetations into one plant functional

type (PFT). This PFT cannot represent the true vegetation composition (shrubs,

sedges, mosses etc.) of peatlands. However, by optimizing the value of $V_{cmax}$ at each

site, we matched simulated GPP with observations so that we had good carbon input

to the soil. The $V_{cmax}$ values estimated in this study ranged from 19 to 89 μmol m$^{-2}$ s$^{-1}$,

these values were not fully comparable with those reported in field studies, or values

which were used in other peatland models because it is more like a representation of

an average of all plants growing in peatland. We found that optimized $V_{cmax}$ had a

significant positive relationship with temperature, and a significant negative

relationship with the latitude of chosen peatland sites location. We speculated that this

can be attributed to 1) length of growing season increases as latitude decreases, and

temperature and incoming solar radiation, increases. Longer growing season may

enhance vegetation productivity (Fang et al., 2003; Nemani et al., 2003; Piao et al.,

2007); 2) with an adequate water supply, leaves open their stomata in response to

warm environments, leading to a higher photosynthetic efficiency (Chapin III et al.,

2011); 3) the influence of temperature on nutrient availability for plants. The

decomposition of plant litter and the release of nitrogen can be enhanced by high

temperature, although litter decomposition is also driven by soil moisture, vegetation

composition, litter quality and their interactions with temperature (Aerts, 2006;





Cornelissen et al., 2007; Gogo et al., 2016). Because nitrogen is one key element in
proteins that are involved in photosynthesis process, photosynthesis capacity is highly
correlated to nitrogen availability (Evans, 1989; Takashima et al., 2004; Walker et al.,
2014). Since the nitrogen cycle is not explicitly included in the ORCHIDEE-PEAT,
the relationship between $V_{cmax}$ and the latitude (and temperature) possibly reflected
the impact of nitrogen on photosynthesis rates.
Previous studies have shown that peatland functioning may have contrasting
responses to variations in water table depth. Among sites incorporated in our study,
Aurela et al. (2007) reported that at the FI-Sii site, drought increased respiration and
thus diminished NEE; Adkinson et al. (2011) reported that reduced water availability
in 2006 constrained photosynthesis capacity at the rich fen CA-Wp3 and consequently
suppressed NEE, while the poor fen CA-Wp2 did not show significant response to the
lower water table; at CA-Wp1 site, Flanagan and Syed (2011) reported that both
photosynthesis and respiration increased in response to the warmer and drier
conditions; Hurkuck et al. (2016) stated that temperature and light played a more
important role than water table depth in controlling respiration and photosynthesis at
DE-Bou site. In field observations, the timing, duration and intensity of drought have
a major impact on the responses of peatland ecosystems. Lund et al. (2012)
demonstrated that at SE-Faj site, a relatively short but severe drought that occurred in
the middle of growing season of 2006 amplified respiration while a long-lasting
drought that occurred at the beginning of growing season of 2008 reduced GPP.
Lafleur et al. (2005) and Sulman et al. (2009) concluded from their studies at CA-Mer
and US-Los that wetter peatlands would show stronger relationship between
respiration and water table than drier peatlands because in a narrow range of the upper
soils, small increases in WT (shallower WT) can result in a large increase in a soil
water content and therefore respiration decrease, while below a critical level, soil
water content shows only small increase with increasing WT and respiration changes
are not so pronounced. Sulman et al. (2010) found that wetter conditions decreased
respiration at fens but increased respiration at bogs, mainly due to different vegetation





composition at these two types of peatland: the fen sites had more shrubs and sedges
while the bog sites had more mosses. In this study, we didn't distinguish between fens
and bogs, and growth of peatland vegetation was not constrained by soil water in the
model, thus the sensitivity of GPP to WT fluctuations in observations was not
included in the model. However, the model can reproduce the pattern that above a
critical level (acrotelm depth), peat respiration decreases with increasing WT
(shallower WT).

ORCHIDEE-PEAT adequately captured the daily, seasonal and across-sites annual

variations in GPP (with $r^2$ = 0.75, 0.86, and 0.93, respectively), but were less able to
reproduce variations in NEE (with $r^2$ = 0.38, 0.61, and 0.27, respectively). One
possible cause is that in the two-layer soil carbon scheme, the dependence of soil
respiration on temperature was parameterized as an exponential function of the soil
layers-weighted average temperature (Text S1), the vertical temperature gradient in
the soil profile was ignored by the model. While field studies have shown that soil
temperature is one of the most important predictors of respiration and values of $Q_{10}$
coefficient depend on the soil depth (Lafleur et al., 2005; D'Angelo et al., 2016).
Another possible cause is that small-scale peatland surface heterogeneities are not
included in the model, which may exert important influences on water and carbon
cycles but has been a challenge for global land surface models (Gong et al., 2013;
Cresto Aleina et al., 2016).

For sites where latent and sensible heat were measured, about half of them used

closed/enclosed path, which may cause an underestimation of LE and H (Twine et al.,
2000). We also need to note that ORCHIDEE-PEAT only diagnose energy fluxes on
one grid-cell and not for each soil tile/PFT present in the same grid cell. A site-varied
and/or time-varied correction of LE and H measurements to force energy balance
closure, and parameterizations of an independent energy budget at peatland may be
helpful for better comparison of simulated and observed energy fluxes at peatland.

**6. Conclusions**





We developed ORCHIDEE-PEAT to simulate soil hydrology and carbon dynamics in
peatlands. The model was evaluated at 30 northern peatland sites (Europe, USA,
Canada, Russia). The optimization of $V_{cmax}$ reduced the errors in carbon simulations,
generally, the model reproduced the spatial gradient and temporal variations in GPP
and NEE well. Water table depth was poorly simulated, possibly due to uncertainties
in water input from non-peatland areas in the grid cell, and lack of representation of
micro-relief, as well as the lack of consideration of the human impacts. A significant
relationship between $V_{cmax}$ and latitude was found, which may be attributed to the
influence of temperature on growing season length and nutrient availability. For NEE
fluxes, improvement brought by forcing the carbon module to use observed WT
values ($WT_{obs}$), instead of calculated by the model, is small, indicating that the
influence of poorly simulated WT on NEE is small.
Our study shows that to reproduce spatial gradients of NEE for northern peatlands,
an average $V_{cmax}$ value is not sufficient. To represent a spatial gradient of carbon
fluxes in large-scale simulations of northern peatlands, incorporating the nitrogen
cycle in peatlands could be helpful, or, an empirical relationship between $V_{cmax}$ and
the latitude (temperature) may be used as a proxy of nitrogen availability. Effects of
water table variations on soil carbon decomposition are modeled as the partitioning of
the acrotelm layer into oxic and anoxic zone, but effects of water table changes on
GPP are not modeled in this study. The model needs further improvement in case to
include the influences of water table on photosynthesis and depth-dependent
influences of soil temperature on soil respiration, as well as an independent energy
budget for peatland in a future model version.




**Competing interests**
The authors declare that they have no conflict of interest




**Code availability**

The code of ORCHIDEE-PEAT will be available upon request. The SVN version of

the code branch is svn://forge.ipsl.jussieu.fr/orchidee/perso/chunjing.qiu/ORCHIDEE,

revision 4596. Please contact the corresponding author to obtain the model.


**Data availability**

Primary data and scripts used in the analysis and other supplementary information can

be obtained from the corresponding author upon request.


*Acknowledgements*

This study was supported by the European Research Council Synergy grant

ERC-2013-SyG-610028 IMBALANCE-P. We would like to thank all the PIs for

giving us permission to use the flux and ancillary data, and all the help and advices

they provided while we were preparing the manuscript. We thank the Polish National

Science Centre which provided funds for site Kopytkowo (PL-Kpt) under projects

UMO-2011/01/B/ST10/07550 and UMO-2015/17/B/ST10/02187, and the Department

of Energy for supporting measurements at Lost Creek fen (US-Los) through the

Ameriflux Network Management Project. We gratefully acknowledge the financial

support provided for La Guette site under the Labex VOLTAIRE

(ANR-10-LABX-100-01) and the PIVOTS project of the R égion Centre – Val de

Loire ((ARD 2020 program and CPER 2015 -2020). Data from the Greenlandic sites

(DK-ZaF and DK-NuF) were provided by the Greenland Ecosystem Monitoring

Programme.




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




**Figures and Tables**



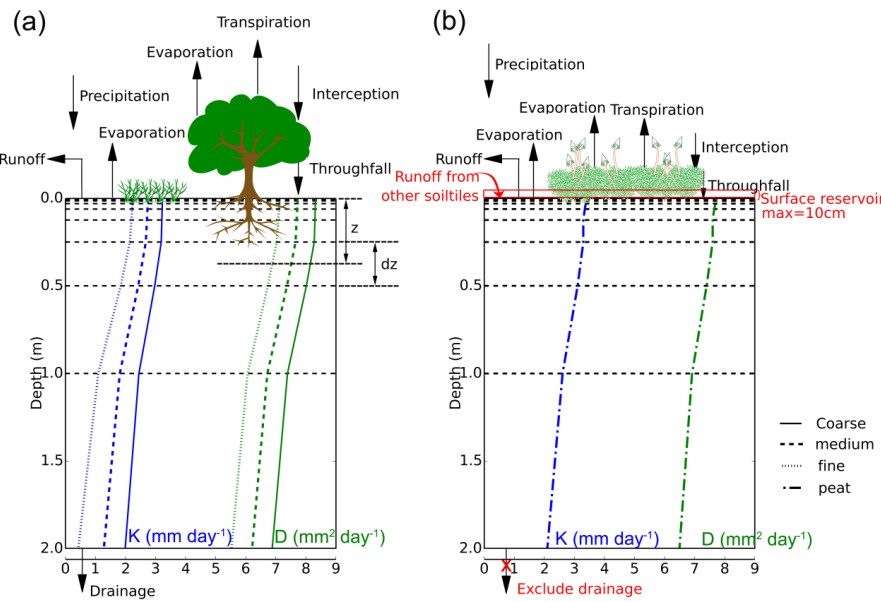


**Fig. 1.** Schematic of the hydrology module in ORCHIDEE. (a) water balance components in
(a) a soil tile with either trees or grasses, (b) a peatland soil tile. Black dashed lines indicate
the position of nodes in the eleven soil layers of the model. Blue lines: vertical profile of
saturated hydraulic conductivity for different soil textures. Green lines: diffusivity for
different soil textures. Vertical axis indicates soil depth, the horizontal axis indicates values
of saturated hydraulic conductivity (K, mm day-1) and diffusivity (D, mm2 day-1), and scales
are logarithmic based 10.








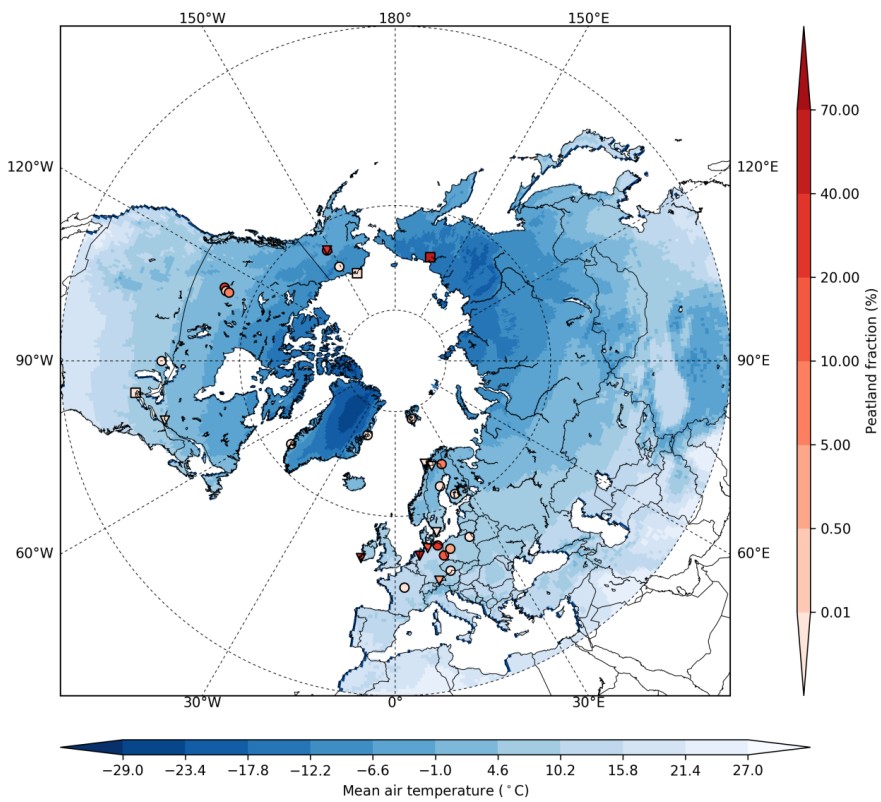


**Fig. 2.** The distribution of 30 peatland sites used in this study. Triangles are bogs; circles are
fens; squares are tundra and marsh. Colors of the markers indicate peatland fractions in the
0.5 ° grid cell. Mean air temperatures is the annual mean from 1999 to 2015, based on the
6-hourly CRU-NCEP 0.5 ° global database.





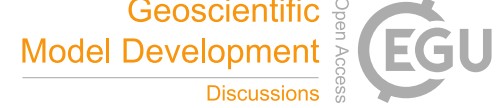



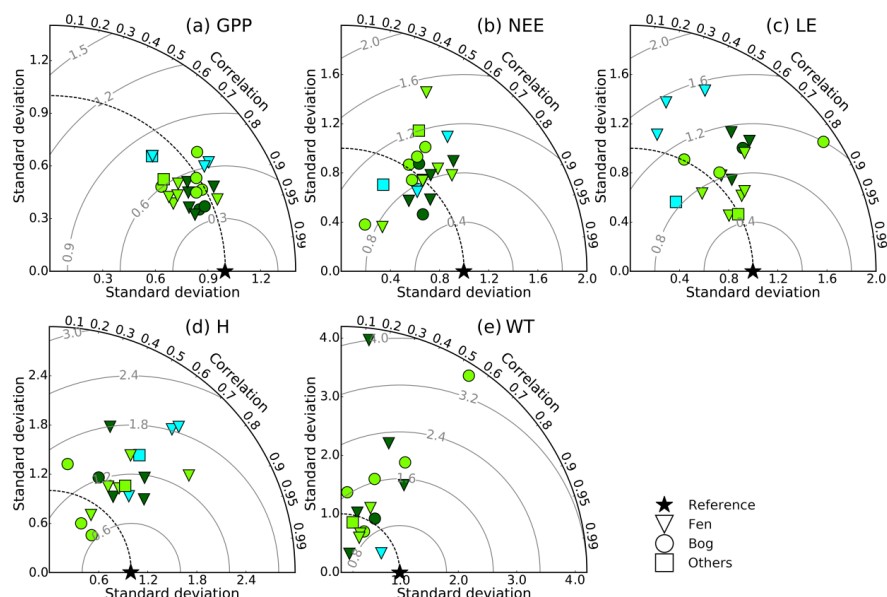

**Fig. 3.** Taylor diagrams of: (a) GPP (g C m-2 day-1); (b) NEE (g C m-2 day-1); (c) LE (W m-2); (d) H (W m-2) and (e) Water table depth (WT, cm). All statistics were calculated using daily averaged data. All points were normalized by dividing the standard deviation of model results by the standard deviation of the corresponding measurement, thus the reference point is 1.0. Light green markers represent temperate sites, dark green markers - boreal sites, blue markers - arctic sites.





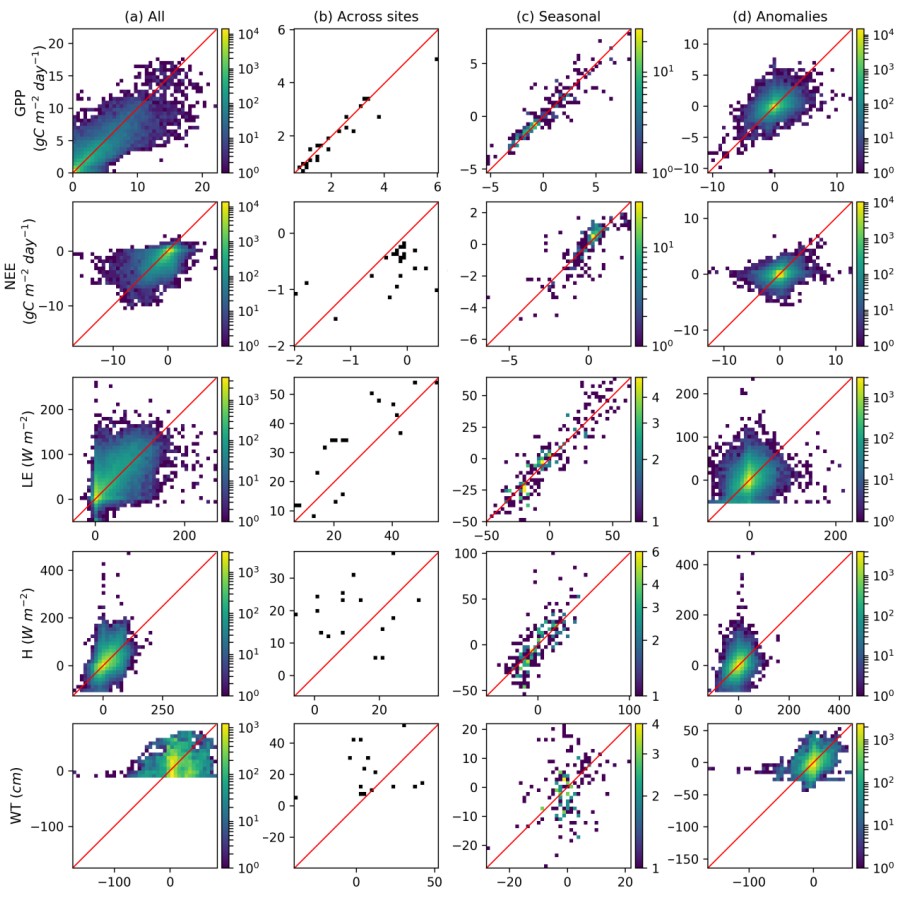

**Fig. 4.** Observed (x-axis) versus simulated (y-axis) fluxes (GPP, NEE, WT, LE, H) at the 22 sites where GPP derived from EC measurements were available. Fluxes were simulated using site-specific optimized Vcmax. The colors of points indicate the number of data in each bin, in panel (b) each data point represents one peatland site. The red line identifies the observations = the simulations.

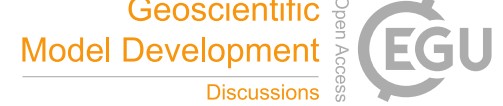

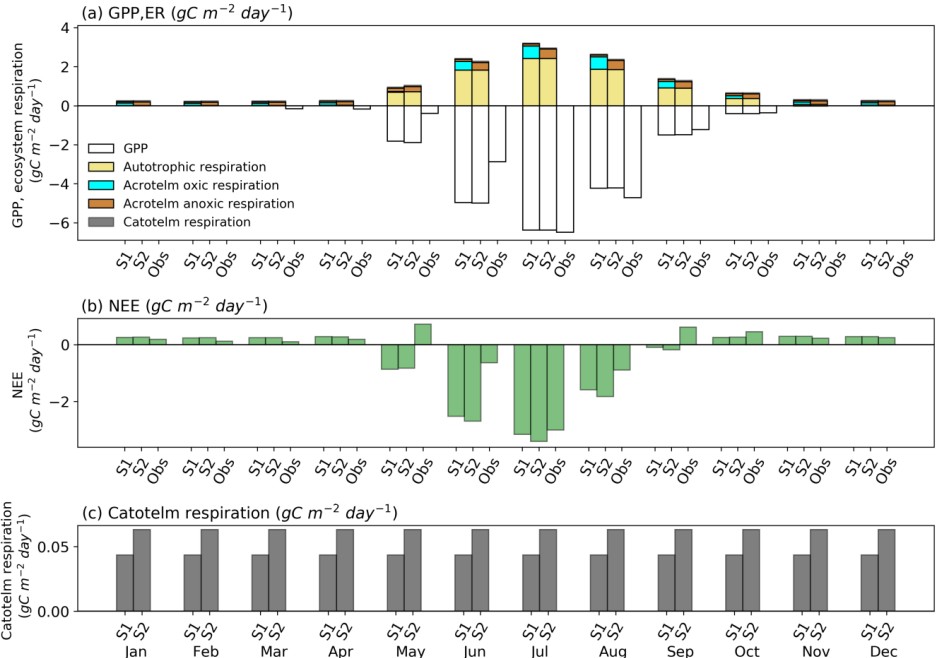

1152
1153

**Fig. 5.** Monthly mean (averaged over 2007–2009) of (a) GPP and ecosystem respiration(ER); (b) NEE; (c) catotelm respiration at Lompolojänkäfen site (FI-Lom). S1: simulated water table (WT) was used in the carbon module; S2: observed WT values (WT$_{obs}$) was used; ob: measured NEE. The graph inserted shows catotelm respiration. By convention, a source of CO2 to the atmosphere is a positive number.






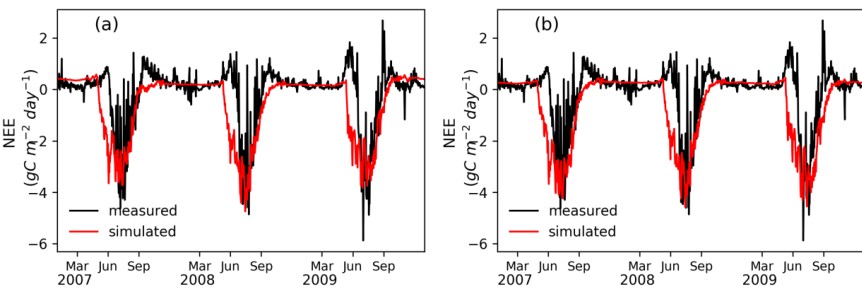

**Fig. 6.** Observed and simulated daily mean NEE at FI-Lom fen site in a) S1 (Simulated WT was used in the carbon module); (b) S2 (modeled water table was assimilated to observed values (WT_{obs}) and was used in the carbon module).





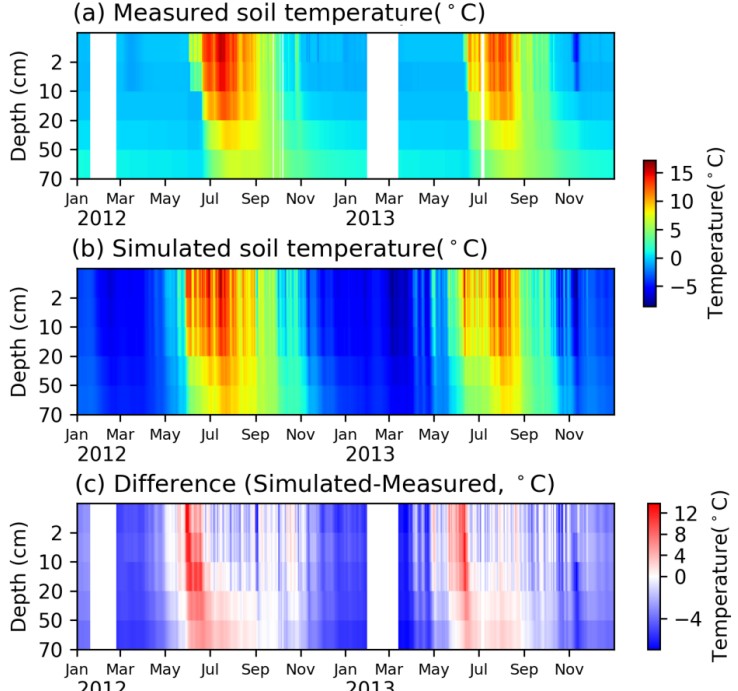

**Fig. 7.** Measured (a), simulated (b) soil temperature, and their difference (c) at DK-Nuf

(64.13 °, -51.39 °) fen site. Soil temperature was measured at 2, 10, 20, 50 and 70 cm below

soil surface. To compare simulated soil temperatures with the measurements, we linearly

interpolated simulated soil temperature in different layers to the depths of the measurements.





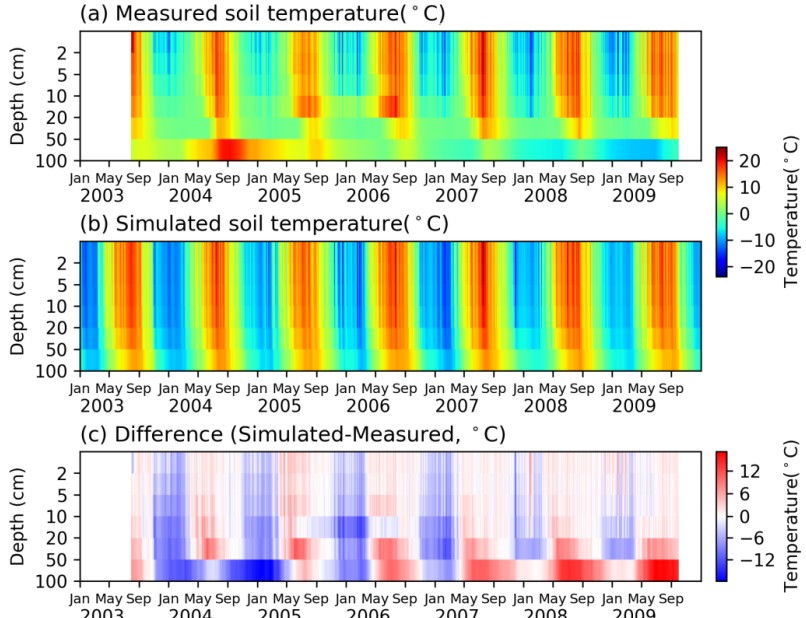

**Fig. 8.** Measured (a), simulated (b) soil temperature, and their difference (c) at CA-Wp1 (54.95°, -112.47°) fen site. The measured soil temperature (a) is the mean of a hummock and a hollow. Soil temperature was measured at 2, 10, 20, 50 and 100 cm below soil surface. To compare simulated soil temperatures with the measurements, we linearly interpolated simulated soil temperature in different layers to the depths of the measurements.





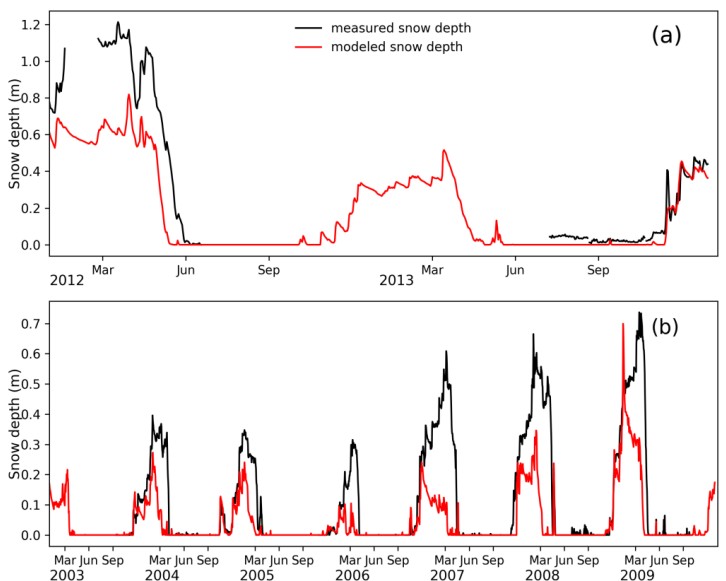


**Fig. 9.** Simulated versus measured snow depth (m) at (a) DK-Nuf fen and (b) CA-Wp1fen.






**Table 1.** Van Genuchten parameters used for different soil texture classes for non-peat soils
(coarse, medium, fine), and for peat. $\theta_s$ is the saturated water content ($m^3\ m^{-3}$), $\theta_r$ is the
residual water content ($m^3\ m^{-3}$); $K_{s\text{-ref}}$ is the reference saturated hydraulic conductivity ($m\ s^{-1}$);
$\alpha$ is the inverse of the air entry suction ($m^{-1}$); $n$ is a dimensionless parameter. In Eq. 1 and Eq.
2, $m = 1-1/n$.

|  | $K_{s\text{-ref}}$ ($m\ s^{-1}$) | $n$ | $\alpha$ ($m^{-1}$) | $\theta_s$ ($m^3\ m^{-3}$) | $\theta_r$ ($m^3\ m^{-3}$) |
|---|---|---|---|---|---|
| COARSE | $1.23\times10^{-5}$ | 1.89 | 7.5 | 0.41 | 0.065 |
| MEDIUM | $2.89\times10^{-6}$ | 1.56 | 3.6 | 0.43 | 0.078 |
| FINE | $7.22\times10^{-7}$ | 1.31 | 1.9 | 0.41 | 0.095 |
| PEAT | $2.45\times10^{-5}$ | 1.38 | 5.07 | 0.90 | 0.15 |





**Table 2.** Sites Characteristics of the 30 peatlands (sites are sorted by latitude from south to north). The first column denotes if the site is used in the second set of simulation (S2, with water table prescribed in the model equal to observed values): y-YES, n-NO. Lat: latitude; Lon: longitude; MAT: long term mean annual air temperature; MAP: long term mean annual precipitation; Peatland fraction: fraction of peatland in the 0.5 ° grid cell which is read from the map of Yu et al. (2010), for cells where there is no peatland, mean fraction (22%) is used. Note that at US-Bog and US-Fen, the precipitation is growing season (from 16th May to 31th August) mean value, thus clarified as 'GS' in the table. Details of S2 and peatland fraction are provided in Sect. 3.3.

| In S2 | Code | Site name | Lat | Lon | climatic zone | Type | MAP (mm) | MAT (°C) | Elevation (m) | Peatland fraction (%) | Period | Citation |
|---|---|---|---|---|---|---|---|---|---|---|---|---|
| n | US-WPT | Winous Point North Marsh | 41.46 | -83.00 | temperate | marsh | 840 | 9.2 | 175 | Mean (22) | 2011-2013 | Chu et al., 2014,2015 |
| n | CA-Mer | Mer Bleue Eastern Peatland | 45.41 | -75.52 | temperate | bog | 944 | 6.0 | 70 | Mean (22) | 1999-2012 | Lafleur et al.,2005 |
| y | US-Los | Lost Creek | 46.08 | -89.98 | temperate | fen | 666 | 3.8 | 470 | Mean (22) | 2000-2010 | Sulman et al., 2009 |
| n | LA-GUE | La Guette peatland | 47.32 | 2.27 | temperate | fen | 880 | 11.0 | 145 | Mean (22) | 2011-2013 | D'Angelo et al.,2016; Laggoun-Défarge et al., 2016 |
| y | DE-Sfn | Schechenfilz Nord | 47.81 | 11.33 | temperate | bog | 1127 | 8.6 | 590 | 3.01 | 2012-2014 | Hommeltenberg et al.,2014 |
| y | CZ-Wet | CZECHWET | 49.02 | 14.77 | temperate | fen | 614 | 7.4 | 426.5 | Mean (22) | 2007-2013 | Dušek et al., 2009 |
| n | DE-Spw | Spreewald | 51.89 | 14.03 | temperate | fen | 559 | 9.5 | 61 | 11.01 | 2010-2014 | |



| y | IE-Kil | Killorglin-Glencar | 51.97 | -9.90 | temperate | bog | 2467 | 10.5 | 150 | 28.97 | 2002-2012 | Sottocornola et al., 2009; McVeigh et al., 2014 |
| y | DE-Bou | Bourtanger Moor | 52.66 | 7.18 | temperate | bog | 799 | 10.0 | 19 | 63.98 | 2011-2014 | Hurkuck et al., 2016 |
| n | PL-Wet | Rzecin wetland | 52.45 | 16.18 | temperate | fen | 526 | 8.5 | 54. | 4.01 | 2006-2013 | Chojnicki et al.,2007; Barabach, 2012; Milecka et al., 2017 |
| n | PL-Kpt | Kopytkowo | 53.59 | 22.89 | temperate | fen | 600 | 7.1 | 109 | Mean (22) | 2013-2015 | Fortuniak et al., 2017 |
| n | DE-Hmm | Himmelmoor | 53.74 | 9.85 | temperate | bog | 838 | 9.0 | 12 | 15.99 | 2012-2014 | Vanselow-Algan et al.,2015 |
| n | DE-Zrk | Zarnekow | 53.88 | 12.89 | temperate | fen | 584 | 8.7 | <0.5 | 23.16 | 2013-2014 | Franz et al., 2016 |
| n | CA-Wp3 | Western Peatland Rich Fen | 54.47 | -113.32 | boreal | fen | 504 | 2.1 | 670 | 29.77 | 2004-2006 | Adkinson et al.,2011 |
| n | CA-Wp1 | Western Peatland | 54.95 | -112.47 | boreal | fen | 504 | 2.1 | 540 | 0.2 | 2003-2009 | Flanagan and Syed, 2011 |
| n | CA-Wp2 | Western Peatland Poor Fen | 55.54 | -112.33 | boreal | fen | 504 | 2.1 | 730 | 8.07 | 2004-2006 | Adkinson et al.,2011 |
| y | SE-faj | Fäjemyr | 56.27 | 13.55 | temperate | bog | 700 | 6.2 | 140 | Mean (22) | 2005-2009 | Lund et al.,2007,2012 |
| n | FI-Sii | Siikaneva | 61.83 | 24.19 | boreal | fen | 713 | 3.3 | 162 | Mean (22) | 2005-2014 | Aurela et al.,2007; Riutta et al., 2007 |
| n | DK-NuF | Nuuk Fen | 64.13 | -51.39 | arctic | fen | 750 | -1.4 | 40 | Mean (22) | 2008-2014 | Westergaard-Nielsen et al., 2013 |



| | | | | | | | | | | | | |
|---|---|---|---|---|---|---|---|---|---|---|---|---|
| y | SE-Deg | Degerö Stormyr | 64.18 | 19.56 | boreal | fen | 523 | 1.2 | 270 | Mean (22) | 2001-2005 | Sagerfors et al.,2008; Nilsson et al.,2008; Peichl et al., 2014 |
| n | US-Bog | Alaska Bog | 64.70 | -148.32 | boreal, thermokarst | bog | 146 (GS) | -2.2 | 100 | 28.01 | 2011-2015 | Euskirchen et al.,2014 |
| n | US-Fen | Alaska Fen | 64.70 | -148.32 | boreal | fen | 146 (GS) | -2.2 | 100 | 28.01 | 2011-2015 | Euskirchen et al.,2014 |
| y | FI-Lom | Lompolojänkkä | 68.00 | 24.21 | boreal | fen | 521 | -1.0 | 269 | 5.08 | 2007-2009 | Aurela et al.,2009 |
| n | SE-Sto (ICOS) | Abisko Stordalen Palsa Bog | 68.36 | 19.05 | boreal, permafrost | bog | 322 | -0.14 | 360 | Mean (22) | 2014-2015 | Malmer et al.,2005; Olefeldt et al.,2012 |
| n | US-Ics | Imnavait Creek Watershed Wet Sedge Tundra | 68.61 | -149.31 | arctic, permafrost | fen | 318 | -7.4 | 920 | Mean (22) | 2007-2011 | Euskirchen et al.,2012, 2016 |
| n | RU-Che | Cherskii | 68.61 | 161.34 | arctic, permafrost | tundra | 200-215 | -12.5 | 4 | 64.09 | 2002-2005 | Corradi et al.,2005; Merbold et al., 2009 |
| n | NO-And | Andøya | 69.14 | 16.02 | boreal | bog | 1060 | 3.6 | 17 | Mean (22) | 2008-2014 | Lund et al.,2015 |
| n | US-Bes | Barrow-Bes (Biocomplexity Experiment South tower) | 71.28 | -156.60 | arctic, permafrost | tundra | 173 | -12 | 4 | Mean (22) | 2005-2008 | Zona et al.,2009 |
| n | DK-Zaf | Zackenberg Fen | 74.48 | -20.55 | arctic, permafrost | fen | 211 | -9.0 | 35 | Mean (22) | 2008-2011 | Stiegler et al.,2016 |
| n | NO-Adv | Adventdalen | 78.19 | 15.92 | arctic, permafrost | fen | 190 | -6.7 | 17 | Mean (22) | 2011-2014 | Pirk et al., 2016 |



*For most of the sites, NEE was partitioned into GPP and ecosystem respiration following the nighttime partitioning method of , except that: NO-And used a light response curve approach following ; CA-Wp1 used the Fluxnet-Canada Research Network (FCRN) standard NEE partitioning procedure following ; and DE-Spw used the online gap filling and flux partitioning tool (http://www.bgc-jena.mpg.de/~MDIwork/eddyproc/) which uses the method proposed by



**Table 3.** Optimized $V_{cmax}$ ($\mu$mol m$^{-2}$ s$^{-1}$) at each site.

| Site | $V_{cmax}$ | Site | $V_{cmax}$ |
|---|---|---|---|
| US-WPT | 80 | FI-Sii | 19 |
| CA-Mer | 25 | DK-NuF | 31 |
| US-Los | 65 | SE-Deg | 23 |
| DE-Sfn | 45 | US-Bog | 42 |
| CZ-Wet | 54 | US-Fen | 56 |
| DE-spw | 89 | FI-Lom | 28 |
| IE-Kil | 28 | RU-che | 35 |
| DE-Bou | 34 | NO-And | 21 |
| DE-Zrk | 33 | DK-Zaf | 37 |
| CA-Wp1 | 38 | NO-Adv | 28 |
| SE-faj | 21 | PL-Kpt | 52 |





**Table 4.** Model performance measures for GPP, WT, NEE, LE and H. The left-hand column shows results with site-specific optimized $V_{cmax}$ at each site, the right-hand column shows results with the fixed $V_{cmax}$ (16 μmol m$^{-2}$ s$^{-1}$) at all sites.

| | Site-specific optimized $V_{cmax}$ | | | | | Default $V_{cmax}$ (constant value, 16 μmol m$^{-2}$ s$^{-1}$) | | | | |
|---|---|---|---|---|---|---|---|---|---|---|
| Flux | RMSD | SDSD | LCS | $r^2$ | MEF | RMSD | SDSD | LCS | $r^2$ | MEF |
| | Overall (Daily variability) | | | | | Overall (Daily variability) | | | | |
| GPP | 1.39 | 0.11 | 1.80 | 0.76 | 0.76 | 3.25 | 6.63 | 0.58 | 0.34 | -0.33 |
| NEE | 1.30 | 0.02 | 1.56 | 0.38 | 0.26 | 1.47 | 1.91 | 0.20 | 0.19 | 0.03 |
| LE | 31.67 | 21.65 | 932.76 | 0.42 | 0.14 | 31.85 | 21.68 | 942.67 | 0.42 | 0.13 |
| H | 35.40 | 96.59 | 1151.28 | 0.24 | -0.50 | 35.42 | 96.77 | 1153.52 | 0.24 | -0.50 |
| WT | 25.93 | 10.26 | 661.80 | 0.01 | -0.56 | 26.21 | 3.54 | 682.21 | 0.02 | -0.59 |
| | Across sites variability | | | | | Across sites variability | | | | |
| GPP | 0.41 | 0.03 | 0.10 | 0.93 | 0.89 | 2.29 | 1.28 | 0.18 | 0.08 | -2.46 |
| NEE | 0.60 | 0.06 | 0.20 | 0.27 | -0.01 | 0.65 | 0.33 | 0.02 | 0.02 | -0.18 |
| LE | 9.85 | 1.13 | 65.49 | 0.71 | 0.50 | 9.78 | 0.90 | 63.89 | 0.71 | 0.51 |
| H | 14.31 | 2.67 | 155.85 | 0.01 | -1.04 | 14.21 | 2.51 | 155.32 | 0.01 | -1.01 |
| WT | 24.40 | 15.20 | 444.83 | 0.02 | -0.82 | 25.39 | 3.57 | 464.55 | 0.04 | -0.97 |
| | Mean seasonal variability | | | | | Mean seasonal variability | | | | |
| GPP | 0.92 | 0.03 | 0.81 | 0.86 | 0.86 | 2.27 | 4.91 | 0.24 | 0.59 | 0.13 |
| NEE | 0.80 | 0.00 | 0.64 | 0.61 | 0.54 | 1.11 | 1.14 | 0.10 | 0.37 | 0.10 |
| LE | 11.49 | 7.75 | 124.23 | 0.83 | 0.78 | 11.79 | 7.63 | 131.38 | 0.82 | 0.77 |
| H | 17.85 | 65.77 | 252.65 | 0.57 | 0.11 | 17.86 | 63.66 | 255.26 | 0.56 | 0.11 |
| WT | 9.87 | 8.32 | 88.88 | 0.06 | -1.38 | 10.00 | 13.84 | 86.04 | 0.11 | -1.44 |
| | Anomalies | | | | | Anomalies | | | | |
| GPP | 1.03 | 0.03 | 1.02 | 0.18 | 0.01 | 1.01 | 0.90 | 0.13 | 0.08 | 0.04 |
| NEE | 0.96 | 0.12 | 0.81 | 0.07 | -0.07 | 0.92 | 0.76 | 0.09 | 0.01 | 0.01 |
| LE | 27.43 | 26.14 | 726.25 | 0.07 | -0.94 | 27.56 | 26.90 | 732.60 | 0.06 | -0.95 |
| H | 28.09 | 81.43 | 707.43 | 0.12 | -1.12 | 28.16 | 83.17 | 709.65 | 0.12 | -1.13 |
| WT | 13.25 | 0.40 | 174.69 | 0.10 | -0.47 | 13.39 | 1.42 | 177.66 | 0.11 | -0.50 |



**Table 5.** Model performance measures of NEE simulations for the site-by-site comparison, the comparison across sites, mean seasonal cycle and anomalies, using modeled (S1) and observed (S2) water table (WT).

| Site | Modeled WT used (S1) | | | | | Observed WT used (S2) | | | | |
|------|------|------|-----|-------|-----|------|------|-----|-------|-----|
| | RMSD | SDSD | LCS | $r^2$ | MEF | RMSD | SDSD | LCS | $r^2$ | MEF |
| CZ-Wet | 2.97 | 3.61 | 4.38 | 0.46 | 0.37 | 2.86 | 3.22 | 4.27 | 0.50 | 0.41 |
| DE-Bou | 1.30 | 0.02 | 1.40 | 0.31 | -0.21 | 1.31 | 0.03 | 1.41 | 0.31 | -0.23 |
| DE-Sfn | 2.98 | 2.98 | 4.27 | 0.20 | 0.02 | 2.98 | 3.08 | 4.15 | 0.21 | 0.02 |
| FI-Lom | 1.05 | 0.01 | 0.94 | 0.46 | 0.21 | 1.08 | 0.02 | 0.95 | 0.49 | 0.16 |
| IE-Kil | 0.48 | 0.000 | 0.16 | 0.29 | -0.37 | 0.49 | 0.002 | 0.16 | 0.32 | -0.44 |
| SE-Deg | 0.64 | 0.03 | 0.33 | 0.51 | 0.09 | 0.57 | 0.01 | 0.29 | 0.51 | 0.26 |
| SE-Faj | 0.65 | 0.01 | 0.33 | 0.31 | -0.36 | 0.65 | 0.02 | 0.33 | 0.32 | -0.39 |
| US-Los | 3.15 | 0.05 | 8.78 | 0.47 | -3.37 | 3.10 | 0.06 | 8.57 | 0.39 | -3.23 |
| Overall | 1.95 | 0.20 | 3.52 | 0.02 | -0.35 | 1.92 | 0.18 | 3.42 | 0.04 | -0.31 |
| Across sites | 0.67 | 0.27 | 0.16 | 0.40 | 0.29 | 0.65 | 0.26 | 0.14 | 0.46 | 0.32 |
| Seasonal | 1.30 | 0.05 | 1.64 | 0.25 | 0.13 | 1.27 | 0.03 | 1.58 | 0.28 | 0.17 |
| Anomalies | 1.18 | 0.22 | 1.17 | 0.003 | -0.34 | 1.17 | 0.21 | 1.17 | 0.001 | -0.33 |





**Table 6.** The results of the ANOVA analysis – the variance of optimized $V_{cmax}$ in relation to chosen variables.

| Variable | F-ratio | p-value | $r^2$ (%) |
|---|---|---|---|
| T | 4.67 | 0.04* | 18.95 |
| P | 0.95 | 0.34 | 4.52 |
| NET_RAD | 0.22 | 0.64 | 1.11 |
| WUE | 0.39 | 0.54 | 1.91 |
| WB | 1.35 | 0.26 | 6.32 |
| LAT | 6.08 | 0.023 * | 23.30 |

* indicates statistical significance at a significance level of 0.05