# Peer review of "ORCHIDEE-PEAT (revision 4596), a model for northern peatland"

_Geoscientific Model Development, 2017_

## Short Comment (SC1) · 10 Oct 2017

Chunjing

As explained in https://www.geoscientific-model-development.net/about/manuscript_types.html GMD is expecting that authors upload the program code of models and the used data sets as a supplement or make the code and data available at a data repository preferable with an associated DOI (digital object identifier) for the exact model version described in the paper. If for some reason your code and/or data cannot be made available in this form as the code availability section in your paper suggests you need to state the reasons why the code is not available or why access is restricted.

[Figure]

Please note that in the code accessibility section you can still point the reader to your web site for updates even if you provide the code as supplement or use a DOI for a release.

All the best Lutz Gross GMD Executive Editor

---

## Referee Comment (RC1) · Anonymous Referee #1 · 20 Oct 2017

This manuscript describes a new peatland model implemented in the ORCHIDEE land model. The model was evaluated by comparing modeled water table, LE, GPP, and NEE to measured eddy covariance fluxes from several peatland field sites. The paper is generally well written and the key processes of the model are clearly described. The introduction section includes a useful review of recent peatland models that does a good job of setting the stage for this model. The paper generally does a good job of identifying uncertainties and potential weaknesses in the model that could be addressed in future work, although I think there is some room for improvement in describing some of these issues in more depth.

[Figure]

I think there are a couple of general areas in which the manuscript could be improved:

1. The key peatland-specific changes to the model are focused on peat carbon pools and hydrology, including a new architecture for simulating peat decomposition using acrotelm and catotelm layers. The modifications to plant processes are less dramatic. In my understanding the model uses an existing C3 grass plant functional type and does not introduce any new peatland-specific vegetation processes. Given the focus of model process changes on decomposition rather than plant processes, it seems strange that the evaluation is so focused on GPP. Why not show and evaluate modeled ecosystem respiration instead of or in addition to GPP? Analyzing respiration fluxes would allow a much better evaluation of the key new model features that are specific to peatland processes. Without an evaluation specific to these new processes, it feels like there is a big piece missing.

2. The approach to optimizing Vcmax is problematic. The optimized site-specific values are compared to a default value that is well outside the range of values that seem to be appropriate for these sites (within the model at least). Figure S3 demonstrates this very clearly for GPP and NEE: the model using the default Vcmax is not even close to reproducing the observed magnitude of photosynthesis at these sites. As a result, the comparison between optimized and default Vcmax simulations is not very informative. It would be more useful if that comparison used the mean or median of the optimized Vcmax values (which is actually used for a different analysis later in the paper). In that case, it would be possible to evaluate whether site-to-site variations in Vcmax were necessary for improving model fidelity. It's not very informative to show that optimized Vcmax is better than a Vcmax that is much too low for every site.

The fact that the default Vcmax based on observations does not work within the model raises further questions. The paper addresses this very briefly (lines 249-251) but I think a more detailed discussion of why the model Vcmax needs to be so much higher than observations would be useful. Were the other photosynthesis-related parameters (LAI, light absorption, etc) in the model consistent with site measurements?

Site-specific optimization of Vcmax could mask other issues with the model, for example underestimates of plant biomass or LAI. I think it would be really helpful to show how modeled LAI compares to measurements, especially among different sites, and whether errors in modeled LAI can explain the latitude/temperature relationship in optimized site Vcmax.

Specific comments:

Lines 173-176: I'm not sure it's that novel that this model is built into a land surface scheme that conserved water, carbon, and energy. Doesn't the LPJ-GUESS model described above have a similar purpose? In any case, if there is not already a peatland submodel built into ORCHIDEE then I wouldn't be that concerned about justifying the purpose of this effort. I think it's clearly valuable to build and evaluate a working peatland submodel within ORCHIDEE.

Line 232: Not all peatlands are grassy. Does this assumption cause issues when applying the model to shrubby or forested peatlands (such as the Old Black Spruce site mentioned a few lines after this)? Were all the peatland sites used for evaluation grassy peatlands?

Line 249-251: It's great that the paper brings up this issue of compensating errors, but it would be better if there were some evaluation of whether the model has systematic errors in LAI, etc.

Line 257: "drainage flux reduced to zero": So there is no water flow out of the peatland unless it is flooded? This seems inconsistent with a lot of real peatland systems.

Line 299-301 and Fig. S1: The difference between the soil carbon dynamics and the peat carbon dynamics is confusing. Do the peat pools contain the Active/Slow/Passive soil carbon pools, or do they replace them? Fig. S1 suggests that all of these pools are present in the peatland (metabolic litter, structural litter, acrotelm, catotelm, active, slow, passive) but this doesn't seem consistent with the description in the text. If the

peat layers are actually replacing the active/slow/passive pools, then Fig. S1 and the text should make that clearer.

Line 308-310: Did this use the observed or simulated water table? How would this be handled in larger-scale or global simulations?

Line 315-316: It would help to show the equation for beta instead of just describing it. Equations for acrotelm height and catotelm depth should also be included.

Is the depth of catotelm and total peat depth calculated? What does the model do if water table goes below the bottom of the peat layer? Can it represent a situation with no catotelm layer? Is there mineral soil beneath the bottom of the peat layers?

Line 331-332: k_A and k_C are defined as fixed parameters, but line 319 says that they have a temperature dependence that is not shown in equations 5-8. These equations should show the complete calculation, including temperature dependence etc.

Line 493-496: If this were the correct explanation, I would expect WT to be more accurately simulated in fens than in bogs. Was that the case?

Line 496-499: This seems like a very likely explanation to me, and something that could be tested by using a range of source-area/peatland-area ratios. Watershed analyses for the sites in question could provide some suggestions of realistic ratios.

Line 515-516: This really highlights how the main peatland-related processes in the model are related to decomposition and respiration, not plant growth. Since that's the case, why is the evaluation so focused on photosynthesis? I think analysis of respiration fluxes would be much more informative, particularly in this case where WT would be expected to have an effect.

Line 531: Water use efficiency doesn't really fit with these other variables. It's a biological parameter, not a climate forcing variable like the other ones.

Line 536-537: It's surprising that there is no difference in Vcmax between fens and

bogs, since those have very different vegetation types and productivities.

Line 540-541: This really seems like it could be compensating for some other error related to vegetation biomass, LAI, or productivity. I would expect higher biomass and LAI in warmer areas, which would drive exactly this type of relationship. I think this should be investigated since the optimization of Vcmax could be masking other important model issues.

Line 549: Why not use this mean value of 40 in the previous comparison, instead of the default value of 16?

Line 560-561: Why are only these two sites discussed and shown in the figure? Was the relevant data not available for other sites, or are these just being used as illustrative examples?

Line 564-566: The suggestion that the issues are due to errors in snow density implies that the snow mass was correct in the model. Is that true?

Line 582-585: Even if optimized Vcmax is an average for the ecosystem rather than a species-specific value, it should be comparable with the observed range among different species that exist in these systems. Other peatland models should definitely be comparable, because any peatland model would be representing an average plant type. I don't think this is a satisfying explanation for not comparing the optimized estimates with measurements. It's just as likely that the model underestimates LAI and needed to tune Vcmax higher to compensate. I don't find any of the three explanation below particularly convincing, and I think bias in LAI or plant biomass is a likely explanation that should be tested.

Line 591-592: Does ORCHIDEE not already take the influence of temperature on photosynthesis into account?

Line 593: If the issue were nutrient availability, I would expect strong contrasts in Vcmax between fen and bog ecosystems, which did not appear to be the case in this study.

[Figure]

Line 603-632: This is a nice review of observed drought effects on peatlands, but the paper doesn't demonstrate whether the model can reproduce any of these effects. Such a demonstration would be very informative.

Line 630-632: It would be better to show that the model reproduces this pattern (in a figure) rather than just asserting that it can.

Line 634-635: If GPP was captured well but NEE was not, then the difference must be due to simulated respiration. This is another case where more analysis of simulated respiration would be very helpful.

Line 666: This implies that water table is not an important feature of carbon cycling according to this model. This seems very inconsistent with the observational literature showing that peatland CO2 fluxes are quite responsive to water table fluctuations (much of which is cited in this manuscript). Some papers have demonstrated that compensating responses of GPP and respiration (e.g. both increasing under a drying trend) can cause NEE to be insensitive to water table fluctuations (e.g. Sulman et al. 2010), but the paper doesn't really demonstrate that the model is reproducing those compensating responses. Given the centrality of water table and hydrology in our understanding of peatland carbon cycling, I think this conclusion that water table isn't actually that important needs to be investigated in more detail, especially in how it affects peat decomposition and ecosystem respiration in the model.

Line 670-671: The paper definitely did not establish that nitrogen availability was the explanation for the latitudinal dependence. It was one of several proposed explanations. In fact, I think it's unlikely to be the explanation because it did not vary consistently with fen/bog type, which is closely related to nitrogen availability.

Table 2: In addition to bog/fen type, it would be informative to include something about the dominant vegetation type (grass, shrub, forested) and maybe aboveground biomass or LAI if available

[Figure]

[Figure]

---

## Referee Comment (RC2) · Anonymous Referee #2 · 21 Oct 2017

The authors present a new peatland model as part of the ORCHIDEE land-surface model. The manuscript is well written and does a nice job of describing recent advances in peatland modeling and identifying the need for the model developments reported here. Specifically, the model simulates water table by prescribing peat-specific hydraulic properties across the 11 soil-profile layers. Water table is then used to determine decomposition rates in in the near-surface acrotelm and deeper, saturated catotelm. The model is evaluated using eddy covariance measurements from 30 sites across northern hemisphere (bog, fen, and tundra). In general, I think the manuscript is in good shape, and I have a few relatively minor comments:

[Figure]

1. Would it be possible for the authors to evaluate model performance of heterotrophic respiration or ER vs. observed values?

2. Line 132 – Should be permafrost "thaw", not "melt

3. Lines 231-232 – While incorporating a peatland-specific PFT is a step in the right direction, I was surprised the authors did not develop a bryophyte or shrub PFT for application in this study, particularly given the range of peatlands used for model comparison. It seems like at the very least, the authors should acknowledge this as a cause of discrepancies between model output and observations.

4. Line 321-324 – Please clarify how the CENTURY-type model of the standard OR-CHIDEE is incorporated in the new decomposition parameterizations for the peatland version. As is, it's not clear how the three-pool set-up relates to these equations.

5. Line 566-567 – The model does incorporate hydraulic properties of peat soils. It seems like it would have been relatively straightforward to also incorporate thermal properties of peats to improve soil temperature performance and its effects on respiration.

6. The authors point toward possible causes of the poor model performance with respect to water table in the Discussion. It would be helpful if they could lay out some practical future steps to improve model performance, particularly given the importance of WT on below-ground C cycling parameters.

---

## Author Comment (AC1) · 14 Dec 2017

**We thank the two anonymous referees very much for their constructive comments. In the following, please find our response to the comments. Our responses are in bold, modifications done in the revised manuscript are in blue. All figure and table numbers, line numbers and pages refer to the initial manuscript version.**

Referee #1

This manuscript describes a new peatland model implemented in the ORCHIDEE land model. The model was evaluated by comparing modeled water table, LE, GPP, and NEE to measured eddy covariance fluxes from several peatland field sites. The paper is generally well written and the key processes of the model are clearly described. The introduction section includes a useful review of recent peatland models that does a good job of setting the stage for this model. The paper generally does a good job of identifying uncertainties and potential weaknesses in the model that could be addressed in future work, although I think there is some room for improvement in describing some of these issues in more depth.

I think there are a couple of general areas in which the manuscript could be improved:
1. The key peatland-specific changes to the model are focused on peat carbon pools and hydrology, including a new architecture for simulating peat decomposition using acrotelm and catotelm layers. The modifications to plant processes are less dramatic. In my understanding the model uses an existing C3 grass plant functional type and does not introduce any new peatland-specific vegetation processes. Given the focus of model process changes on decomposition rather than plant processes, it seems strange that the evaluation is so focused on GPP. Why not show and evaluate modeled ecosystem respiration instead of or in addition to GPP? Analyzing respiration fluxes would allow a much better evaluation of the key new model features that are specific to peatland processes. Without an evaluation specific to these new processes, it feels like there is a big piece missing.

**While our initial focus was on peatland productivity and carbon intake, of course GPP and ER are linked. So we followed the reviewer's suggestion to incorporate an analysis of ecosystem respiration. To do so, we added analyses and discussion of simulated vs. measured ecosystem respiration. In the first set of simulation (S1) in which the modeled water table were used in the carbon module, with the site-specific $V_{cmax}$, the model showed good performance in capturing both spatial and temporal variations in ER, with $r^2$ of 0.78, 0.89, 0.86 for daily variations, across-sites annual variations and seasonal variations, respectively, and MEF of 0.75, 0.79, and 0.86, respectively. These results were compared with simulations using a fixed $V_{cmax}$ (the mean of the optimized $V_{cmax}$, 40 $\mu$mol m$^{-2}$ s$^{-1}$), as suggested by the review in the second comment. We conclude that by**

**taking site-to-site variations in $V_{cmax}$ into consideration, model performances for carbon fluxes (GPP, ER and NEE) were improved. Table4 and Figure4 have been updated to include results of ER and results with the mean of the optimized $V_{cmax}$, and the description of the results from Line474 to Line487 was rephrased as:** "For the 22 sites where NEE and ER measurements were available, the errors in the three carbon fluxes－GPP, ER, NEE were significantly reduced by optimizing $V_{cmax}$ at each site (Table 4, Fig. 4, Fig. S4). With site-specific $V_{cmax}$ values (Site-by-site model performances are shown in Fig. S5 to S10 in Supplementary Materials), the overall (all the daily data from all the 22 sites) performance of the model was good for GPP ($r^2 = 0.76$, MEF = 0.76), ER ($r^2 = 0.78$, MEF = 0.75), and acceptable for NEE ($r^2 = 0.38$, MEF = 0.26) (Fig. 4, Table 4). Seasonal variations in carbon fluxes were well captured by the model ($r^2 = 0.61$ to 0.86). The spatial across-sites gradients of annual mean GPP and ER were generally good, with $r^2$ of 0.93 and 0.89, and lower for NEE ($r^2 = 0.27$). Compared to simulations with a fixed $V_{cmax}$ (the mean of the optimized values of 40 µmol m$^{-2}$ s$^{-1}$), there were large improvements in capturing spatial gradients of carbon fluxes ($r^2$ increased from 0.20 to 0.93, from 0.27 to 0.89 and from 0.16 to 0.27 for GPP, ER and NEE, respectively, while the RMSD reduced by 63%, 48%, and 9%). This result indicates that model-data disagreement can be largely reduced by using site-specific $V_{cmax}$ instead of a fixed (mean) value. In future regional simulations, spatial variations in $V_{cmax}$ should be taken into account. There was, however, no significant improvement in LE, H and WT by using site-specific $V_{cmax}$ values (Table4)." **We also compared simulated ER of S1 with the second set of simulation (S2, in which the measured water table was used) with the ER observations: the model showed only a small improvement in reproducing ER when $WT_{obs}$ was used (Table 5 was added to show the results). Fig.S6 was added to show simulated vs. measured ER at each site.**

**Table 5.** Model performance measures of ER simulations for the site-by-site comparison, the comparison across sites, mean seasonal cycle and anomalies, using modeled (S1) and observed (S2) water table (WT). SDSD and LCS are two signals discriminated from the mean squared deviation, see Sect. 3.4.

| Site | Modeled WT used (S1) | | | | | Observed WT used (S2) | | | | |
|------|------|------|------|-------|------|------|------|------|-------|------|
|      | RMSD | SDSD | LCS | $r^2$ | MEF | RMSD | SDSD | LCS | $r^2$ | MEF |
| CZ-Wet | 1.45 | 0.86 | 0.87 | 0.81 | 0.68 | 1.51 | 1.05 | 0.79 | 0.81 | 0.66 |
| DE-Bou | 0.78 | 0.03 | 0.50 | 0.69 | 0.64 | 0.77 | 0.03 | 0.50 | 0.69 | 0.65 |
| DE-Sfn | 0.96 | 0.10 | 0.79 | 0.61 | 0.59 | 0.97 | 0.09 | 0.82 | 0.60 | 0.58 |
| FI-Lom | 0.46 | 0.00 | 0.19 | 0.85 | 0.84 | 0.45 | 0.02 | 0.18 | 0.85 | 0.84 |
| IE-Kil | 0.44 | 0.01 | 0.01 | 0.09 | 0.51 | 0.42 | 0.01 | 0.01 | 0.13 | 0.48 |
| SE-Deg | 0.69 | 0.26 | 0.19 | 0.75 | 0.62 | 0.64 | 0.16 | 0.23 | 0.75 | 0.68 |
| SE-Faj | 0.58 | 0.07 | 0.08 | 0.87 | 0.60 | 0.59 | 0.08 | 0.07 | 0.88 | 0.59 |
| US-Los | 0.63 | 0.01 | 0.39 | 0.85 | 0.85 | 0.60 | 0.00 | 0.35 | 0.87 | 0.87 |
| Overall | 0.79 | 0.09 | 0.51 | 0.78 | 0.76 | 0.79 | 0.09 | 0.51 | 0.78 | 0.76 |
| Across sites | 0.31 | 0.01 | 0.06 | 0.82 | 0.76 | 0.32 | 0.01 | 0.06 | 0.82 | 0.74 |
| Seasonal | 0.45 | 0.06 | 0.15 | 0.91 | 0.89 | 0.44 | 0.07 | 0.13 | 0.92 | 0.89 |
| Anomalies | 0.62 | 0.07 | 0.31 | 0.21 | 0.19 | 0.63 | 0.08 | 0.31 | 0.20 | 0.17 |

2. The approach to optimizing Vcmax is problematic. The optimized site-specific values are compared to a default value that is well outside the range of values that seem to be appropriate for these sites (within the model at least). Figure S3 demonstrates this very clearly for GPP and NEE: the model using the default Vcmax is not even close to reproducing the observed magnitude of photosynthesis at these sites. As a result, the comparison between optimized and default Vcmax simulations is not very informative. It would be more useful if that comparison used the mean or median of the optimized Vcmax values (which is actually used for a different analysis later in the paper). In that case, it would be possible to evaluate whether site-to-site variations in Vcmax were necessary for improving model fidelity. It's not very informative to show that optimized Vcmax is better than a Vcmax that is much too low for every site.

The fact that the default Vcmax based on observations does not work within the model raises further questions. The paper addresses this very briefly (lines 249-251) but I think a more detailed discussion of why the model Vcmax needs to be so much higher than observations would be useful. Were the other photosynthesis-related parameters (LAI, light absorption, etc) in the model consistent with site measurements? Site-specific optimization of Vcmax could mask other issues with the model, for example underestimates of plant biomass or LAI. I think it would be really helpful to show how modeled LAI compares to measurements, especially among different sites, and whether errors in modeled LAI can explain the latitude/temperature relationship in optimized site Vcmax.

**The reviewer raises a fair point that a comparison between the optimized and the default $V_{cmax}$ value (16 μmol m$^{-2}$ s$^{-1}$) is not as informative as it could be in this study. The default value applied by Largeron et al. (2017, Geosci. Model Dev. Discuss.) was derived for three low productivity sites. When this value was applied at our dataset, GPP and NEE were underestimated. Thus, to make a more apples-to-apples test, we added a comparison between optimized and the mean of the optimized $V_{cmax}$ values (40 μmol m$^{-2}$ s$^{-1}$), as suggested by the reviewer. The comparison to the default $V_{cmax}$ is removed from the manuscript. The description of the results from Line474 to Line487 was rephrased, as it is mentioned in our response to the first comment of the reviewer.**

**Our use of site optimized $V_{cmax}$ is one way to account for large variance in a key ecosystem parameter. There is a large reported variation of $V_{cmax}$ in observations. For instance, $V_{cmax}$ value for Sphagnum at the Old Black Spruce site in Canada were 5, 14 and 6 μmol m$^{-2}$ s$^{-1}$ during spring, summer and autumn respectively, while that for Pleurozium were 7, 5, and 7 μmol m$^{-2}$ s$^{-1}$ (Williams and Flanagan, 1998, PCE); Bubier et al. (2011, Oecologia) reported that $V_{cmax}$ for three ericaceous shrubs (*Vaccinium myrtilloides*, *Ledum groenlandicum* and *Chamaedaphne calyculata*) at Mer Bleue bog in Canada ranged from 67 to 137 μmol m$^{-2}$ s$^{-1}$ among the control and four nutrient addition treatments (measured $V_{cmax}$ for the three shrubs in the control plots are 84.6 ± 13.5 μmol m$^{-2}$ s$^{-1}$, 78.1 ± 13.4 μmol m$^{-2}$ s$^{-1}$, and 132.1 ± 31.2 μmol m$^{-2}$ s$^{-1}$, respectively); The $V_{cmax}$ value applied by the McGill wetland model for evergreen shrubs is 17 μmol m$^{-2}$ s$^{-1}$, which is the median value of over 50 measurements for *Chamaedaphne calyculata* and *Ledum groenlandicum* (St-Hilaire et al., 2010, Biogeosciences). Wu et al. (2016, Geosci. Model Dev.) used values of 60, 50, 40μmol m$^{-2}$ s$^{-1}$ for evergreen shrubs,**

deciduous shrubs and sedges respectively. The optimized model $V_{cmax}$ values in our study ranged from 19 to 89 μmol m$^{-2}$ s$^{-1}$ (the mean value is 40 μmol m$^{-2}$ s$^{-1}$), considering that the model optimized value represents an average for the ecosystem, we argue that the model value is not substantially above observations or values used in other land surface models.

We agree with the reviewer that site-specific optimization of $V_{cmax}$ could compensate for biases in LAI, plant biomass, etc. Unfortunately, at most of the sites, LAI was measured or estimated (by optical in-situ methods, annual litter collection, or from remote sensing) only once during the periods in question. We have an available time-series of measured LAI at IE-Kil – see Fig.S1(a), LAI was overestimated by the model at IE-Kil. Fig.S1 (b) showed that LAI was overestimated at sites with low reported (measured or estimated) LAI and underestimated at sites with higher reported values. As for aboveground biomass, there was no systematic error among sites. We emphasized the bias in LAI in the text, on Page16, Line463: "……, with a mean value of 40 μmol m$^{-2}$ s$^{-1}$. The calibration of $V_{cmax}$ may compensate for biases in other model parameters. A brief comparison between simulated and reported (measured/estimated) LAI and aboveground biomass showed that there are no systematic errors (Fig. S1).".

[Figure]

**Fig. S1.** (a) Simulated vs. measured leaf area index (LAI) at the blanket bog IE-Kil, Ireland. (b) Simulated vs. reported (measured/estimated) LAI across peatland sites, dashed line is a hypothetical 1:1 regression line. Note that in (b), the reported LAI was estimated at some sites. (c) Simulated vs. measured aboveground biomass, assuming that the carbon content of dry biomass is 50%.

Specific comments:
Lines 173-176: I'm not sure it's that novel that this model is built into a land surface scheme that conserved water, carbon, and energy. Doesn't the LPJ-GUESS model described above have a similar purpose? In any case, if there is not already a peatland submodel built into ORCHIDEE then I wouldn't be that concerned about justifying the purpose of this effort. I think it's clearly valuable to build and evaluate a working peatland submodel within ORCHIDEE.

The reviewer is right, the LPJ-GUESS does describe a similar development, however, there is no water input from surrounding areas (Chaudhary et al., 2016, Biogeosciences), so conservation is scale-dependent. We rephrased the sentences on Page6, Line173 as follows : "This new peat model is incorporated consistently into the land surface scheme in order to conserve water, carbon and energy at scales going from local sites to grid-based

large-scale applications in an Earth System Modeling context.**"**

Line 232: Not all peatlands are grassy. Does this assumption cause issues when applying the model to shrubby or forested peatlands (such as the Old Black Spruce site mentioned a few lines after this)? Were all the peatland sites used for evaluation grassy peatlands?
**The sites used for evaluation include grassy, shrubby, and forested peatlands (Table 2). We note the possible discrepancies between model output and observations in the text as suggested by Reviewer#2. Please refer to our response to the third comment of Reviewer#2 (Lines 231-232).**

Line 249-251: It's great that the paper brings up this issue of compensating errors, but it would be better if there were some evaluation of whether the model has systematic errors in LAI, etc.
**As shown in Fig. S1b, LAI was overestimated at sites with low reported LAI and underestimated at sites with high reported values, there was no systematic error in LAI.**

Line 257: "drainage flux reduced to zero": So there is no water flow out of the peatland unless it is flooded? This seems inconsistent with a lot of real peatland systems.
**We would like to note that although we considered deep drainage from peatland as negligible due to the low permeability of the catotelm (Ingram et al., 1978, EJSS; Rezanezhad et al., 2016, Chem. Geol. ), the waterflow out of the peatland (as runoff) occurs not only when the peatland is flooded. In ORCHIDEE, the partitioning between water infiltration and surface runoff is computed through a time-splitting procedure (d'Orgeval, 2006, PhD thesis), with the maximum infiltration rates described as an exponential probability density distribution. The infiltration-excess water creates runoff. Thus in the model, the infiltration excess water will first fills the above-surface water reservoir, and then leaves the grid cell as runoff.**
**To clarify this, we added these sentences in the text, Page9, Line259: "**……an above surface water reservoir with a maximum height of 10 cm was added (Fig. 1b). In the model, the partitioning between water infiltration and surface runoff is computed through a time-splitting procedure, with the maximum infiltration rates described as an exponential probability density distribution (d'Orgeval, 2006, Diss. Paris). The infiltration-excess water of peatland first fills the above-surface water reservoir, then leaves the grid cell as runoff. Water in this above-surface reservoir re-infiltrates into the peat soil on the next time step (Largeron et al., 2017).**".**

Line 299-301 and Fig. S1: The difference between the soil carbon dynamics and the peat carbon dynamics is confusing. Do the peat pools contain the Active/Slow/Passive soil carbon pools, or do they replace them? Fig. S1 suggests that all of these pools are present in the peatland (metabolic litter, structural litter, acrotelm, catotelm, active, slow, passive) but this doesn't seem consistent with the description in the text. If the peat layers are actually replacing the active/slow/passive pools, then Fig. S1 and the text should make that clearer.
**The reviewer is right, the description of the carbon module is not clear enough. We improved the description in the text on Page10, Line295 to "**…...Decomposed litter carbon

from these two pools is then distributed into three soil carbon pools: the active, slow and passive pool, similar to the CENTURY model (Parton et al., 1988). Both temperature and moisture functions are used to control soil carbon decomposition rates (Text S1). In ORCHIDEE-PEAT, these standard processes are kept the same as in Krinner et al. (2005) for non-peatland vegetation (Fig. S2, black dashed box). For the peatland vegetation, we added a peat carbon module, in which the three soil carbon pools (active, slow, passive) are replaced by two pools forming distinct layers, following Kleinen et al. (2012) (Fig. S2, red dashed box)." **and we modified Fig. S2 as follows:**

[Figure]

**Fig. S2.** Schematic overview of litter and soil carbon dynamics in ORCHIDEE-PEAT. For non-peatland vegetation (the black dashed box), decompositions of carbon in the two litter pools and three soil pools, and carbon flows between them are adapted from the CENTURY model (Parton et al., 1988); for peatland vegetation (the red dashed box), the active, slow and passive soil carbon pools are replaced by a two-layered model, following Kleinen et al. (2012).

Line 308-310: Did this use the observed or simulated water table? How would this be handled in larger-scale or global simulations?

**The simulated mean summer minimum water table position ($WT_{min}$) over the observational period is used here. $WT_{min}$ was derived from a 'preparation run (S0)'. Specifically, we first ran the model at each site using the same simulation protocol as described in Sect. 3.3, but with peat carbon module deactivated. Then $WT_{min}$ can be diagnosed from the output of this simulation (S0) and be fed into the model in S1 and S2.**

**We explained this procedure in the text on Page10, Line308 to: "……we used the average of simulated minimum summer water table position ($WT_{min}$) over the observational period to demarcate the boundary between the acrotelm and the catotelm at each site to take into**

account local site conditions. We conducted a "preparation run (S0)", in which the model was run at each site using the same protocol (Sect. 3.3), but with the peat carbon module deactivated. $WT_{min}$ was diagnosed from the output of S0 before feeding into the peat carbon module in S1 and S2 (Sect. 3.3). Soil carbon exerts no feedback effects on the soil temperature and hydraulics in the structure of our model, thus S0 and S1 produce the same simulated water table.". In large-scale or global simulations, we can either conduct the same "preparation run" or set $WT_{min}$ to a constant value, for example, Wania et al. (2009, Global Biogeochem. Cycles) and Spahni et al. (2013, Clim. Past.) used 0.3 m as the interface between the acrotelm and the catotelm.

Line 315-316: It would help to show the equation for beta instead of just describing it. Equations for acrotelm height and catotelm depth should also be included. Is the depth of catotelm and total peat depth calculated? What does the model do if water table goes below the bottom of the peat layer? Can it represent a situation with no catotelm layer? Is there mineral soil beneath the bottom of the peat layers?
We added equations of beta (Eq. 9) and acrotelm depth (Eq.10) on Page11, Line325:

$$\beta = \begin{cases} \beta = 1, & WT_{min} - WT \leq 0 \\ \beta = \dfrac{H_A - (WT_{min} - WT)}{H_A}, & 0 < WT_{min} - WT < H_A \\ \beta = 0, & WT_{min} - WT \geq H_A \end{cases}, \tag{9}$$

$$H_A = \frac{C_A}{\rho_A \cdot C_{f,A}}, \tag{10}$$

The depth of catotelm can be calculated using carbon fraction in the catotelm and the catotelm density, as in Kleinen et al. (2012, Biogeosciences). However, since the initiation and climate history of each site are unknown, we assumed that all sites initiated 10100 years ago, with a constant present-day climate condition since their initiation and the peatland area hasn't changed, thus the simulated peat depth can't be compared to the measured depth.
The model was started with no catotelm layer, the carbon started to accumulate in the acrotelm layer, and as soon as carbon occurred in the acrotelm layer, a prescribed fraction of the acrotelm carbon was moved to the catotelm. When simulated water table (WT) drops below the acrotelm ($WT_{min}$), the whole acrotelm layer is supposed to decompose aerobically, as shown by Eq. 9, while the whole catotelm layer is still decomposing anaerobically. In the hydrology module, the total soil depth is 2m, we assumed that all layers in the peat soil profile hold peat-specific hydraulic properties, and there is no mineral soil beneath the peat soil. While the soil thermodynamics in the soil thermal module has 32 layers (38m), in which the top first 11 layers are identical to layers in hydrology, soil profiles in one grid cell are treated as mineral soil, and the dominant texture is used to define soil thermal properties.

Line 331-332: k_A and k_C are defined as fixed parameters, but line 319 says that they have a temperature dependence that is not shown in equations 5-8. These equations should show

the complete calculation, including temperature dependence etc.

**We revised the equations on Page11, Line 318: "……**Decomposition of peat carbon is controlled by temperature ($f_T$) and parameterized as an exponential function:

$f_T = Q_{10}\exp((T-T_{ref})/10\,°C)$ with $Q_{10} = 2.0$ and $T_{ref} = 30\,°C$ (Text S1). Soil carbon fluxes are given by:

$$F_{AC} = k_p f_T C_A \quad ,$$ (5)

$$R_{A,o} = \beta k_A f_T C_A \quad ,$$ (6)

$$R_{A,a} = (1 - \beta) v k_A f_T C_A \quad ,$$ (7)

$$R_C = k_C f_T C_C \quad ,$$ (8)

Line 493-496: If this were the correct explanation, I would expect WT to be more accurately simulated in fens than in bogs. Was that the case?

**We can't conclude that WT should be more accurately simulated in fens than bogs because we don't know the real amount of water input from non-peatland areas to peatland at fen sites. In this study, we routed all runoff from non-peatland soils into peatland. Considering that water table is relatively sensitive to the peatland area fraction in the grid cell (Fig. S11), it's hard to quantify if this water input setup caused greater errors in bogs than fens or not. The Taylor diagram (Fig. 3f) showed that there is no significant evidence for concluding that WT of fens are better simulated than bogs. We added a sentence on Page17, Line496 to point out the possible cause of this result: "……**an extra water source for bogs than only rainfall. However, the model did not perform better for fens (Fig. 3f), possibly because the amount of water that was routed into the fen was in error**.".**

[Figure]

**Fig. S11.** Sensitivity test of simulated water table to peatland area fraction in the grid cell, performed at the fen site FI-Lom.

Line 496-499: This seems like a very likely explanation to me, and something that could be tested by using a range of source-area/peatland-area ratios. Watershed analyses for the sites in question could provide some suggestions of realistic ratios.

**We agree with the reviewer that watershed analyses could be helpful, but we feel that it's out the scope of this study. It could be considered for further developments of the model. Here, we performed a sensitivity test of simulated water table to peatland area fraction in the grid cell at one fen site (FI-Lom) to show the dependence of simulated water table on peatland area fraction (Fig. S11). We point out the dependence in the text, Page17, Line496: "……3) WT depends on water input from surrounding non-peatland areas: the greater the peatland fraction in the grid cell, the smaller runoff input from other soils to the peatland, hence resulting in a deeper water table in the peatland (Fig. S11). ".**

Line 515-516: This really highlights how the main peatland-related processes in the model are related to decomposition and respiration, not plant growth. Since that's the case, why is the evaluation so focused on photosynthesis? I think analysis of respiration fluxes would be much more informative, particularly in this case where WT would be expected to have an effect.

**As NEE is the small residual of GPP and ER, wrong values of GPP could be one of major sources for errors in simulated NEE, especially when we only have one PFT to represents peatland vegetations. We agree with the reviewers that respiration fluxes are informative, thus we added analyses of ecosystem respiration. Please refer to our response to the first comment of the reviewer.**

Line 531: Water use efficiency doesn't really fit with these other variables. It's a biological parameter, not a climate forcing variable like the other ones.

**Water use efficiency and water balance were included because we would like to find a**

variable / parameter that is possibly related to the optimized $V_{cmax}$, and may be used in the future to prescribe the spatial pattern of $V_{cmax}$ in larger scale simulations in the future. So here we included not only climate forcing variables, but also these two biological parameters. We rephrased the name of Fig.S5 as: "**Fig. S15.** Relationship between optimized $V_{cmax}$ and meteorological variables and biological parameters, as well as latitude of the sites location"

Line 536-537: It's surprising that there is no difference in Vcmax between fens and bogs, since those have very different vegetation types and productivities.

**We recognize that bogs are precipitation-fed and nutrient-poor while fens are fed by precipitation and groundwater and can be either oligotrophic or eutrophic. However, previous studies have shown that along a bog-rich fen gradient in Alberta, Canada, the total above-ground net primary production exhibited a pattern of bog < poor fen < wooded moderate-rich fen> extreme rich fen> sedge fen (Szumigalski and Bayley, 1996, Wetlands), the productivity of the bog was not significantly lower than the poor fen and was even higher than the sedge and the extreme-rich fen. Also in Alberta, Thormann and Bayley (1997, Ecoscience) compared total aboveground plant production along a bog-fen-marsh gradient in Alberta, Canada, and found that the bog and the three fens (a lacustrine sedge fen, a riverine sedge fen and a floating sedge fen) had a similar NPP, the lacustrine sedge fen was even significantly less productive than the bog. The sites used in our study include wooded fens, wooded bogs, grassy fens and grassy bogs. Among them, we can't see a significant difference in dominant vegetation types between fens and bogs (we don't know relative abundances of grasses vs. shrubs vs. trees at each site though). We compared measured GPP of fens with that of bogs, there is no significant difference between them (P=0.63), as shown in the figure below.**

| Site | Type | Aboveground biomass (kg/m2) | Dominant vegetation type |
|------|------|------------------------------|--------------------------|
| DE-Bou | bog | grass dominated: 0.577; heather and moss dominated: 0.517; mixed: 0.303 | grasses, mosses |
| SE-Faj | bog | shrubs: 0.153; graminoids: 0.077; moses: 0.192 | shrubs,grasses,mosses |
| CA-Mer | bog | vascular: 0.356; mosses: 0.144 | shrubs,mosses |
| NO-And | bog | | shrubs, grasses,mosses |
| DE-Sfn | bog | | trees,shrubs,grasses,mosses |
| US-Bog | bog | | trees,mosses |
| SE-Deg | fen | vascular:0.049; mosses:0.065 | shrubs,grasses,mosses |
| CA-Wp3 | fen | 0.157 | grasses,mosses |
| CA-Wp2 | fen | 0.231 | shrubs,grasses,mosses |
| DK-Zaf | fen | 0.471 | grasses,mosses |
| CZ-Wet | fen | 0.57 | grasses |
| NO-Adv | fen | 0.85 | shrubs, grasses,mosses |
| CA-Wp1 | fen | 1.08 | trees,shrubs,mosses |
| US-Los | fen | 1.336 | trees,shrubs,grasses |

| DE-Spw | fen | trees |
| PL-Kpt | fen | grasses,reeds and ferns |
| DE-Zrk | fen | grasses |
| DK-NuF | fen | grasses,mosses |
| US-Fen | fen | grasses,forbs |
| FI-Sii | fen | shrubs,grasses,mosses |
| FI-Lom | fen | shrubs,grasses,mosses |

[Figure]

Line 540-541: This really seems like it could be compensating for some other error related to vegetation biomass, LAI, or productivity. I would expect higher biomass and LAI in warmer areas, which would drive exactly this type of relationship. I think this should be investigated since the optimization of Vcmax could be masking other important model issues.

**The measured LAI indeed is larger in warmer areas, but we would like to mention that there is no systematic bias in LAI or biomass, as shown in Fig.S1. Verheije et al. (2013, Biogeosciences) demonstrated that Earth system models could be improved by taking plant traits variations within PFTs into account, and proposed relationships between trait parameters and the climate, which can be used to define the parameter values for each grid cell. Considering that there is no available observational-based trait-climate relationships that can be used for peatland vegetations, we optimized $V_{cmax}$ at each site and built the relationship between the optimized $V_{cmax}$ and the latitude (temperature), which showed better performance than using a mean value. The peat PFT in our study represents an average of the ecosystem, not a specific plant type. A broad decrease of $V_{cmax}$ with latitude in the northern hemisphere has also been documented by Walker et al. (2017, New Phytologist), assuming that $V_{cmax}$ was constrained by the rate of N uptake, with the rate of N uptake calculated as a function of soil C, N and mean annual air temperature. We note this in the text on Page20, Line587: "……relationship with the** latitude of chosen peatland sites location. A decrease of $V_{cmax}$ with latitude in the northern hemisphere, like the one inferred from optimized sites values, has also been documented by

Walker et al. (2017), who assumed that $V_{cmax}$ was constrained by the rate of N uptake, with the rate of N uptake calculated as a function of soil C, N and mean annual air temperature. We speculate the dependence of optimized $V_{cmax}$ on latitude found in Sect. 4.2 can be attributed to……**".**

Line 549: Why not use this mean value of 40 in the previous comparison, instead of the default value of 16?
**The mean value of 40μmol m$^{-2}$ s$^{-1}$ is used in the revised manuscript. Please refer to our responses to the second comment.**

Line 560-561: Why are only these two sites discussed and shown in the figure? Was the relevant data not available for other sites, or are these just being used as illustrative examples?
**We have data for other sites. The underestimation of soil temperature in winter and overestimation in summer occurred at most of these sites. DK-Nuf and CA-Wp1 are just used as illustrative examples. We corrected the text on Page19, Line560: "**……soil temperature was underestimated in winter and overestimated in summer by our model (Fig. 7 and 8, results from sites DK-Nuf and CA-Wp1 are shown as illustrative examples).**"**

Line 564-566: The suggestion that the issues are due to errors in snow density implies that the snow mass was correct in the model. Is that true?
**We didn't validate the simulated snow mass because of lack of available data. We rephrased the text on Page19, Line564: "**……can be caused by the bias in snow processes of the model, such as underestimation of snow mass, and/or overestimation of snow density and…**".**

Line 582-585: Even if optimized Vcmax is an average for the ecosystem rather than a species-specific value, it should be comparable with the observed range among different species that exist in these systems. Other peatland models should definitely be comparable, because any peatland model would be representing an average plant type. I don't think this is a satisfying explanation for not comparing the optimized estimates with measurements. It's just as likely that the model underestimates LAI and needed to tune Vcmax higher to compensate. I don't find any of the three explaination below particularly convincing, and I think bias in LAI or plant biomass is a likely explanation that should be tested.
**The reviewer is right, the optimized $V_{cmax}$ should be compared with the observed range among different species. Therefore we added these sentences on Page20, Line582:** "……The $V_{cmax}$ values estimated in this study ranged from 19 to 89 μmol m$^{-2}$ s$^{-1}$, with a mean value of 40 μmol m$^{-2}$ s$^{-1}$. These values were not fully comparable with values reported for a specific vegetation type, as they are averages for all plants growing in the peatland ecosystem. As stated in Sect. 2.2, observed $V_{cmax}$ varies strongly among different species and sites. $V_{cmax}$ of mosses at the Old Black Spruce site (Canada) varied from 5 to 14 μmol m$^{-2}$ s$^{-1}$ (Williams and Flanagan, 1998), In a nutrient addition experiments conducted by Bubier et al. (2011), $V_{cmax}$ for ericaceous shrubs in a temperate bog ranged from 67 to 137μmol m$^{-2}$ s$^{-1}$, with $V_{cmax}$ for *Vaccinium myrtilloides*, *Ledum groenlandicum* and *Chamaedaphne calyculata* valued at 84.6 ±13.5 μmol m$^{-2}$ s$^{-1}$, 78.1 ±13.4 μmol m$^{-2}$ s$^{-1}$, and 132.1 ±31.2 μmol m$^{-2}$ s$^{-1}$ in the plots

with no nutrient addition. The optimized model $V_{cmax}$ in our study was within the range of these observations. Meanwhile, the values we inferred from sites to match peak GPP are comparable to those used in other land surface models: the McGill wetland model used a value of 17 μmol m$^{-2}$ s$^{-1}$ for evergreen shrubs (St-Hilaire et al., 2010); the CLASS-CTEM model (Wu et al., 2016) used 60, 50, 40 μmol m$^{-2}$ s$^{-1}$ for evergreen shrubs, deciduous shrubs and sedges, respectively; the values for mosses in these two models were adapted from the study of Williams and Flanagan (1998). **".**

Line 591-592: Does ORCHIDEE not already take the influence of temperature on photosynthesis into account?
**ORCHIDEE does take the influence of temperature on photosynthesis into account by parameterizing the temperature dependences of Michaelis-Menten constants, CO$_2$ compensation point following Medlyn et al. (2002, Plant, cell & environment). And temperature acclimation of photosynthesis rates constants is included in ORCHIDEE following Yin et al. (2009, NJAS-Wageningen J. Life Sci.). We thus removed the following sentences on Page20, line 591-592: "……2) with an adequate water supply, leaves open their stomata in response to warm environments, leading to a higher photosynthetic efficiency (Chapin III et al., 2011);".**

Line 593: If the issue were nutrient availability, I would expect strong contrasts in Vcmax between fen and bog ecosystems, which did not appear to be the case in this study.
**As we mentioned above, the sites used in this study include wooded fens, wooded bogs, grassy fens and grassy bogs, among them, there is no significant difference in dominant vegetation types between fens and bogs. Meanwhile, there is neither significant difference in measured biomass between fens and bogs (P=0.097) nor significant difference in measured GPP (P=0.63).**

Line 603-632: This is a nice review of observed drought effects on peatlands, but the paper doesn't demonstrate whether the model can reproduce any of these effects. Such a demonstration would be very informative.
**We added these sentences to demonstrate results of the model on Page22, Line628:** "……and growth of peatland vegetation was not constrained by water table depth in the model. Therefore, the sensitivity of GPP to WT fluctuations in observations was not included in the model. As a consequence, the model neither captured the reported decrease of photosynthesis due to drought at CA-Wp3 (Adkinson et al., 2011) and SE-Faj (Lund et al., 2012), nor the increase of photosynthesis as a result of lower water table at CA-Wp1 (Flanagan and Syed, 2011). However, the model can reproduce the pattern that above a critical level (acrotelm depth), peat respiration decreases with increasing WT (Fig.5, Fig.S13), as reported at site CA-Mer and US-Los (Lafleur et al., 2005; Sulman et al., 2009). **".**

Line 630-632: It would be better to show that the model reproduces this pattern (in a figure) rather than just asserting that it can.
**The decrease of soil respiration with increasing WT (shallower) was shown in Fig.5 and Fig. S13. We added this sentence on Page22, Line630: "**…… The model can reproduce the

pattern that above a critical level (acrotelm depth), peat respiration decreases with increasing WT (Fig.5, Fig.S13), as reported at site CA-Mer and US-Los (Lafleur et al., 2005; Sulman et al., 2009).**".**

Line 634-635: If GPP was captured well but NEE was not, then the difference must be due to simulated respiration. This is another case where more analysis of simulated respiration would be very helpful.

**Ecosystem respiration was relatively well captured by the model. We added these sentences on Page22, Line634: "**……variations in GPP (with $r^2$ = 0.75, 0.86, and 0.93, respectively) and ER (with $r^2$=0.78, 0.86, and 0.89, respectively), but were less able to reproduce variations in NEE (with $r^2$ = 0.38, 0.61, and 0.27, respectively)**.** Note that in the two-layer soil carbon scheme, the dependence of soil respiration on temperature was parameterized as an exponential function of the soil layers-weighted average temperature (Text S1). ……and values of $Q_{10}$ coefficient depend on the soil depth (Lafleur et al., 2005; D'Angelo et al., 2016). Small-scale peatland surface heterogeneities are not included in the model,**"**

Line 666: This implies that water table is not an important feature of carbon cycling according to this model. This seems very inconsistent with the observational literature showing that peatland CO2 fluxes are quite responsive to water table fluctuations (much of which is cited in this manuscript). Some papers have demonstrated that compensating responses of GPP and respiration (e.g. both increasing under a drying trend) can cause NEE to be insensitive to water table fluctuations (e.g. Sulman et al. 2010), but the paper doesn't really demonstrate that the model is reproducing those compensating responses. Given the centrality of water table and hydrology in our understanding of peatland carbon cycling, I think this conclusion that water table isn't actually that important needs to be investigated in more detail, especially in how it affects peat decomposition and ecosystem respiration in the model.

**The point we were trying to make here is that although water table was poorly simulated by the model, it was good enough to simulate ER (NEE) properly. With water table being forced to be equal to observed values in S2, there were no large improvements in simulated ER, NEE (Table5, Table6, Fig. S13). This is because the oxic decomposition in the acrotelm (β), which is the main component of soil respiration, was calculated by comparing the height of the acrotelm with the WT depth, though absolute values of water table depth in S1 and S2 were quite different (Fig. S8), β were not so different. We took Lompolojänkkäfen site (FI-Lom) as an example, in which WT was most severely underestimated. As shown by Fig. S12, difference between β of S1 and S2 only occurred during short periods and mainly in winter when decompositions were inhibited by the low temperature. We performed an additional simulation (S3), in which we assumed that water table was more severely underestimated by the model (water table used in S3 was consistently 20cm deeper than in S1), thus the acrotelm was more exposed to the air in S3 (Fig. S12). S3 showed much larger ecosystem respiration and hence smaller carbon sequestration than S1. We clarified this by added these sentences on Page18, Line524: "**……an overestimation (more negative values) of NEE in the warm period (May-September). The influence of WT on respiration was parameterized as the

separation of oxic (β in Eq. 6) vs. anoxic (1-β in Eq. 7) decomposition in the acrotelm. Although absolute values of simulated WT in S1 and WT$_{obs}$ in S2 were quite different (Fig. S8), the values of β were not very different (Fig.S12). Therefore the simulated WT was good enough to properly replicate ER (Fig.S13). An additional simulation (S3) performed at FI-Lom showed that if WT was more severely underestimated, e.g. WT in S3 was consistently 20 cm deeper than in S1, the acrotelm was exposed to oxygen for longer time, resulting in larger ER and hence smaller carbon sequestration in S3 (Fig.S12, Fig.S13).**". We rephrased the sentences in abstract on Page3, Line105: "**……likely due to the uncertain water input to the peat from surrounding areas. However, the poor performance of WT did not greatly affect predictions of ER and NEE.**", and the sentences in conclusion on Page23, Line665: "**……instead of calculated by the model, was small, indicating that the simulated WT was reliable to predict ER and NEE properly.**"

[Figure]

**Fig. S12.** The fraction of the acrotelm where carbon decomposes under oxic conditions (β) at Lompolojänkkä fen site (FI-Lom). S1: simulated water table (WT) were used in the carbon module; S2: observed water table (WT$_{obs}$) were used in the carbon module; S3: assumed that water table were 20cm deeper than simulated results, thus (WT−20cm) were used in the carbon module.

[Figure]

**Fig. S13.** Cumulative ER (left figure) and NEE (right figure) at Lompolojänkkä fen site (FI-Lom). S1: simulated water table (WT) were used in the carbon module; S2: observed water table ($WT_{obs}$) were used in the carbon module; S3: assumed that water table were 20cm deeper than simulated results, thus (WT−20cm) were used in the carbon module.

Line 670-671: The paper definitely did not establish that nitrogen availability was the explanation for the latitudinal dependence. It was one of several proposed explanations. In fact, I think it's unlikely to be the explanation because it did not vary consistently with fen/bog type, which is closely related to nitrogen availability.

**Not all fens in this study are nutrient rich, for example, SE-Deg (Peichl et al., 2014, Environ. Res. Lett.), FI-Sii (Aurela et al., 2007, Tellus), CA-Wp2 (Adkinson et al., 2011, J. Geophys. Res. Biogeosciences) are oligotrophic fens, thus there is a large variation in $V_{cmax}$ of fens. And there is no significant difference in biomass, GPP between fens and bogs. Meanwhile, Walker et al. (2017, New Phytologist) found that $V_{cmax}$ values decreased with latitude in the northern hemisphere if the rate of nitrogen uptake was parameterized as a function of soil C, N, and mean annual air temperature. Thus, we can't rule out the possibility that the relationship was caused by nitrogen availability.**

Table 2: In addition to bog/fen type, it would be informative to include something about the dominant vegetation type (grass, shrub, forested) and maybe aboveground biomass or LAI if available

**We included the dominant vegetation type and LAI, and aboveground biomass in the Table2, detailed description of the sites can be found in the supplement material.**

---

## Author Comment (AC2) · 14 Dec 2017

**We thank the two anonymous referees very much for their constructive comments. In the following, please find our response to the comments. Our responses are in bold, modifications done in the revised manuscript are in blue. All figure and table numbers, line numbers and pages refer to the initial manuscript version.**

Referee #2

The authors present a new peatland model as part of the ORCHIDEE land-surface model. The manuscript is well written and does a nice job of describing recent advances in peatland modeling and identifying the need for the model developments reported here. Specifically, the model simulates water table by prescribing peat-specific hydraulic properties across the 11 soil-profile layers. Water table is then used to determine decomposition rates in in the near-surface acrotelm and deeper, saturated catotelm. The model is evaluated using eddy covariance measurements from 30 sites across northern hemisphere (bog, fen, and tundra). In general, I think the manuscript is in good shape, and I have a few relatively minor comments:

1. Would it be possible for the authors to evaluate model performance of heterotrophic respiration or ER vs. observed values?
**We added comparisons of simulated vs. observed ER, please refer to our response to the first comment of Reviewer#1.**

2. Line 132 – Should be permafrost "thaw", not "melt
**Corrected now in the text**.

3. Lines 231-232 – While incorporating a peatland-specific PFT is a step in the right direction, I was surprised the authors did not develop a bryophyte or shrub PFT for application in this study, particularly given the range of peatlands used for model comparison. It seems like at the very least, the authors should acknowledge this as a cause of discrepancies between model output and observations.
**Currently, ORCHIDEE (both the standard ORCHIDEE and ORCHIDEE-PEAT) lacks representation of mosses and shrubs. In the grid-based simulations, we do not know fractional coverage of the peatland vegetation at each site. Wania et al. (2009, Global Biogeochem. Cy.) parameterized flood-tolerant C3 graminoids and *Sphagnum* in LPJ-WHy to represent peatland-specific vegetations, with peatland extent defined from an organic soil map and the fractional cover of PFTs determined by bioclimatic conditions including temperature, water table depth, inundation stress etc. Stocker et al. (2014, Geosci. Model Dev.) applied a version of Wania et al's model but removed the**

upper temperature limitation of the peatland-specific PFTs and further included three additional PFTs — flood tolerance C4 grasses, tropical evergreen and tropical raingreen tree PFTs, with peatland extent diagnosed by TOPMODEL. Previous studies have shown that there was considerable overlap between the plant traits ranges among different plant functional types, while variations in plant traits within PFTs can be even greater than the difference in means among PFTs (Verheije et al., 2013, Biogeosciences; Wright et al., 2005, New Phytol; Laughlin et al., 2010, Funct. Ecol.). For simplicity, in this study, we applied only one PFT to represent an average of all vegetations growing in the peatland ecosystem. However, only one key photosynthetic parameter—$V_{cmax}$ of the PFT has been tuned to match with observations at each studying sites, other processes and parameters of this PFT was inherited from a C3 grass, this simplification may cause discrepancies between model outputs and observations.

Druel et al. (2017, Geosci. Model Dev. Discuss.) added non-vascular plants (bryophytes and lichens), boreal grasses, and shrubs into ORC-HL-VEGv1.0, biogeochemical and biophysical processes of these new PFTs were defined and evaluated in their study. Their work is in parallel with our model, after both ORCHIDEE-PEAT and ORC-HL-VEGv1.0 are incorporated into the main branch of ORCHIDEE in the future, it will then be possible to verify how many plant functional types are needed by the model to reliably simulate the peatlands at site-level and larger scales, though the vegetations implemented by Druel et al. are not peatland-specific. To acknowledge these, we added these sentences on Page8, Line 230: "……and extensive root systems (Boutin and Keddy, 1993; Iversen et al., 2015). Previous peatland models have incorporated more than one PFT to represent peatland plants and dynamically simulate fractional vegetation cover. For example Wania et al. (2009b) separated flood-tolerant C3 graminoids and *Sphagnum* moss in LPJ-WHy to represent peatland-specific vegetation, with peatland extent defined from an organic soil map and the fractional cover of PFTs determined by bioclimatic conditions including temperature, water table depth, inundation stress etc. Stocker et al. (2014) applied a version of this model but removed the upper temperature limitation of the peatland-specific PFTs and further included three additional PFTs — flood tolerant C4 grasses, tropical evergreen and tropical raingreen tree PFTs, with peatland extent diagnosed by the TOPMODEL scheme. At present, however, ORCHIDEE-PEAT lacks representation of dynamic moss and shrub covers, and we do not know the fractional coverage of different vegetation types at each site in grid-based simulations. Previous studies have shown that there was considerable overlap between the plant traits ranges among different plant functional types, while variations in plant traits within PFTs can be even greater than the difference in means among PFTs (Verheijen et al., 2013; Wright et al., 2005; Laughlin et al., 2010). Therefore, for simplicity, we applied the PFT of C3-grass with a shallower rooting depth to represent the average of vegetation growing in northern peatlands.

Only one key photosynthetic parameter—$V_{cmax}$ of this PFT has been tuned to match with observations at each site. This simplification may cause discrepancies between model output and observations. Druel et al. (2017) added non-vascular plants (bryophytes and lichens), boreal grasses, and shrubs into ORC-HL-VEGv1.0. Their work is in parallel with our model and will be incorporated into the model in the future. It will then be possible to verify how many plant functional types are needed by the model to reliably simulate the peatlands at

site-level and larger scale.**".**

4. Line 321-324 – Please clarify how the CENTURY-type model of the standard ORCHIDEE is incorporated in the new decomposition parameterizations for the peatland version. As is, it's not clear how the three-pool set-up relates to these equations.
**We clarified the structure of the carbon module in ORCHIDEE-PEAT in the text and modified Fig.S1 to show the scheme of the model clearer, please refer to our response to Reviewer#1 (Specific comments, Line 299-301 and Fig. S1) for details.**

5. Line 566-567 – The model does incorporate hydraulic properties of peat soils. It seems like it would have been relatively straightforward to also incorporate thermal properties of peats to improve soil temperature performance and its effects on respiration.
**ORCHIDEE-PEAT lacks parameterization of peat-specific thermal characteristics due to the original thermal scheme of the model. Within a gridcell, different soil columns are represented but only the charactericstic of the dominant are used to define the thermal properties (soil thermal conductivity and heat capacity) in the model. The model configuration doesn't allow us to assign different properties for each soil column in the same one grid cell. An ideal solution would be to change the structure of the model so that peat soil can have peat-specific thermal properties while non-peat soil columns keep using the dominant mineral soil texture. This is the approach we used for soil hydraulics. We would like to mention that a study by Guimberteau et al. (2017, Geosci. Model Dev.) conducted in parallel to our study added the feedback effects of soil organic carbon concentration on soil thermics into ORCHIDEE, specifically, soil physical properties of one grid cell is a weighted average of mineral soil and organic soil, with carbon content for organic soil derived from the soil organic carbon map from NCSCD. This approach takes thermal properties of peat (pure organic soil) into account in a simplified way. Guimberteau et al.'s development can be used by ORCHIDEE-PEAT after the model is merged into the main branch of ORCHIDEE in the near future.**

6. The authors point toward possible causes of the poor model performance with respect to water table in the Discussion. It would be helpful if they could lay out some practical future steps to improve model performance, particularly given the importance of WT on below-ground C cycling parameters.

**We added following seteneces to the discussion, Page22, Line641: "……**depend on the soil depth (Lafleur et al., 2005; D'Angelo et al., 2016). Correct representation of peatland hydrology is a challenging problem in large-scale land surface models (Wania et al., 2009a; Wu et al., 2016). The simulated water table by ORCHIDEE-PEAT depends on water inflows from the surrounding non-peatland areas, and a water routing analysis on sub-grid scales can be included to improve the model performance for water table in the future (Ringeval et al., 2012; Stocker et al., 2014). Other studies have shown that microtopography exerts important influences on hydrological dynamics of peatlands, however, to capture the influence of microtopography on water table, high-resolution micro-topographic feature and vegetation information are needed (Gong et al., 2013; Shi et al., 2015). **"** .

---

## Author Comment (AC3) · 14 Dec 2017

We thank the executive editor. We modified the Code and data availability section as below:

Code availability: The access of the source code is available online via the following address: (http://forge.ipsl.jussieu.fr/orchidee/browser/perso/chunjing.qiu/ORCHIDEE), but its access is restricted. Readers interested in running the model should follow the instructions at http://orchidee.ipsl.fr/index.php/you-orchidee, and contact the corresponding author for a username and password. Data availability: Measured Eddy Covariance fluxes and related meteorological data can be obtained from the FLUXNET

database (http://fluxnet.ornl.gov/), the Ameriflux database (http://ameriflux.lbl.gov/), and from investigators upon request. Model outputs are available at: https://files.lsce.ipsl.fr/public.php?service=files&t=c12c831ef46cd2bf6d1f61b6e65f8c98.
* * *